# A disease associated mutant reveals how Ltv1 orchestrates RP assembly and rRNA folding of the small ribosomal subunit head

Ebba K. Blomqvist[1], Haina Huang[1,2¤], Katrin Karbstein[1,2]*

**1** Department of Integrative Structural and Computational Biology, The Herbert Wertheim UF Scripps Institute for Biomedical Innovation & Technology, Jupiter, Florida, United States of America, **2** The Skaggs Graduate School of Chemical and Biological Sciences, The Scripps Research Institute, La Jolla, California, United States of America

¤ Current address: Arrakis Therapeutics, Waltham, Massachusetts, United States of America

* katrin.karbstein@ufl.edu

**Data Availability Statement:** All data are contained within the manuscript or via ProteomeXchange with identifier PXD046239.

## Abstract

Ribosomes are complex macromolecules assembled from 4 rRNAs and 79 ribosomal proteins (RPs). Their assembly is organized in a highly hierarchical manner, which is thought to avoid dead-end pathways, thereby enabling efficient assembly of ribosomes in the large quantities needed for healthy cellular growth. Moreover, hierarchical assembly also can help ensure that each RP is included in the mature ribosome. Nonetheless, how this hierarchy is achieved remains unknown, beyond the examples that depend on direct RP-RP interactions, which account for only a fraction of the observed dependencies. Using assembly of the small subunit head and a disease-associated mutation in the assembly factor Ltv1 as a model system, we dissect here how the hierarchy in RP binding is constructed. A combination of data from yeast genetics, mass spectrometry, DMS probing and biochemical experiments demonstrate that the LIPHAK-disease-associated Ltv1 mutation leads to global defects in head assembly, which are explained by direct binding of Ltv1 to 5 out of 15 RPs, and indirect effects that affect 4 additional RPs. These indirect effects are mediated by conformational transitions in the nascent subunit that are regulated by Ltv1. Mechanistically, Ltv1 aids the recruitment of some RPs via direct protein-protein interactions, but surprisingly also delays the recruitment of other RPs. Delayed binding of key RPs also delays the acquisition of RNA structure that is stabilized by these proteins. Finally, our data also indicate direct roles for Ltv1 in chaperoning the folding of a key rRNA structural element, the three-helix junction j34-35-38. Thus, Ltv1 plays critical roles in organizing the order of both RP binding to rRNA and rRNA folding, thereby enabling efficient 40S subunit assembly.

## Author summary

Ribosomes make proteins in all cells. During that process, they are responsible not just for the correct translation of the genetic code into amino acids, but also for selecting the correct mRNAs for translation. Finally, damage in mRNAs requires active translation.

**Funding:** This work was supported by National Institute of Health grants R35-GM136323 and HHMI Faculty Scholar Grant 55108536 (to K.K.). The funders had no role in study design, data collection and analysis, decision to publish, or preparation of the manuscript.

**Competing interests:** The authors have declared that no competing interests exist.'

Changes in ribosome numbers or ribosome composition can perturb each of these functions and can lead to diseases. Thus, a better understanding of how cells ensure efficient and correct ribosome assembly is paramount. Using a mutation that is associated with a dermatological condition, LIPHAK syndrome, we describe here novel roles for the assembly factor Ltv1 in orchestrating the assembly of RPs into the head of the small ribosomal subunit. Our experiments, a combination of mass spectrometry, RNA structure probing, genetics and biochemical assays, demonstrate that Ltv1 helps recruit some RPs, while delaying the incorporation of others, thereby establishing the hierarchy of RP binding. In addition, our results also demonstrate roles for Ltv1 in modulating rRNA folding. Finally, the data establish the identity of a previously described quality control step that is bypassed by a cancer-associated mutation.

## Introduction

In all forms of life, ribosomes are responsible not just for producing the right amount of the correct protein, but also the detection of damaged mRNAs. Both functions depend on the functional integrity of the ribosome pool, as well as efficient ribosome assembly, as reduced ribosome numbers can affect both mRNA selection [1,2], as well as its surveillance for damage [3].

In eukaryotes, ribosomes are assembled from their constituent 4 ribosomal RNAs (rRNAs) and 79 ribosomal proteins (RPs) with the help of ~ 200 assembly factors (AFs), which are largely conserved from yeast to humans. These factors integrate rRNA processing with binding of RPs, chaperone rRNA folding, allow for regulation, prevent premature translation initiation, and facilitate quality control [4–6].

The binding of ribosomal proteins (RPs) to ribosomal RNA (rRNA) is very hierarchical, as first characterized by Nomura and Nierhaus [7–9]. A potential advantage of this hierarchy is the ability to ensure that every RP is incorporated into the nascent ribosome, as the next step cannot proceed without the previous one. This is critical, as demonstrated by findings that individual RPs are often depleted in cancer cells, which is associated with a poor prognosis [10–13]. In addition, hierarchical binding can help avoid dead-end pathways, which limit the efficiency of assembly.

The small subunit head is constructed from the 18S rRNA 3'-domain and is the last of the subunit substructures to emerge. Because of its relatively late assembly, individual assembly steps can be identified using cryo-electron microscopy (cryo-EM, [14–19]), as well as biochemical investigations [20–25], rendering the head a model system to study rRNA folding [22] and RP binding *in vivo* [20–24,26]. Its assembly is initiated co-transcriptionally with binding of Rps5 (uS7), Rps16 (uS9), Rps18 (uS13) and Rps28 [27], and then continues with binding of Rps12, Rps15 (uS19), Rps17, Rps19, Rps25 [19]. The final proteins to assemble onto the head are Rps3 (uS3), Rps10, Rps20 (uS10), Rps29 (uS14), Rps31 and Asc1 [25]. While these three groups essentially recapitulate Nomura's description of early, middle and late binding proteins [8], a more detailed analysis of the available structural evidence indicates a more intricate assembly pathway with additional hierarchies (Fig 1A) that is not captured by Nomura's original map.

Importantly, RP-induced rRNA folding [28–30] or contacts between an earlier and a later-binding RP can explain some of the hierarchy, e.g., the dependence of Rps19 binding on prior binding of Rps16 and Rps18 (Fig 1A). Additionally, in some cases an assembly factor blocks binding of an RP: *e.g.*, Utp15, a component of the UtpA complex, blocks premature

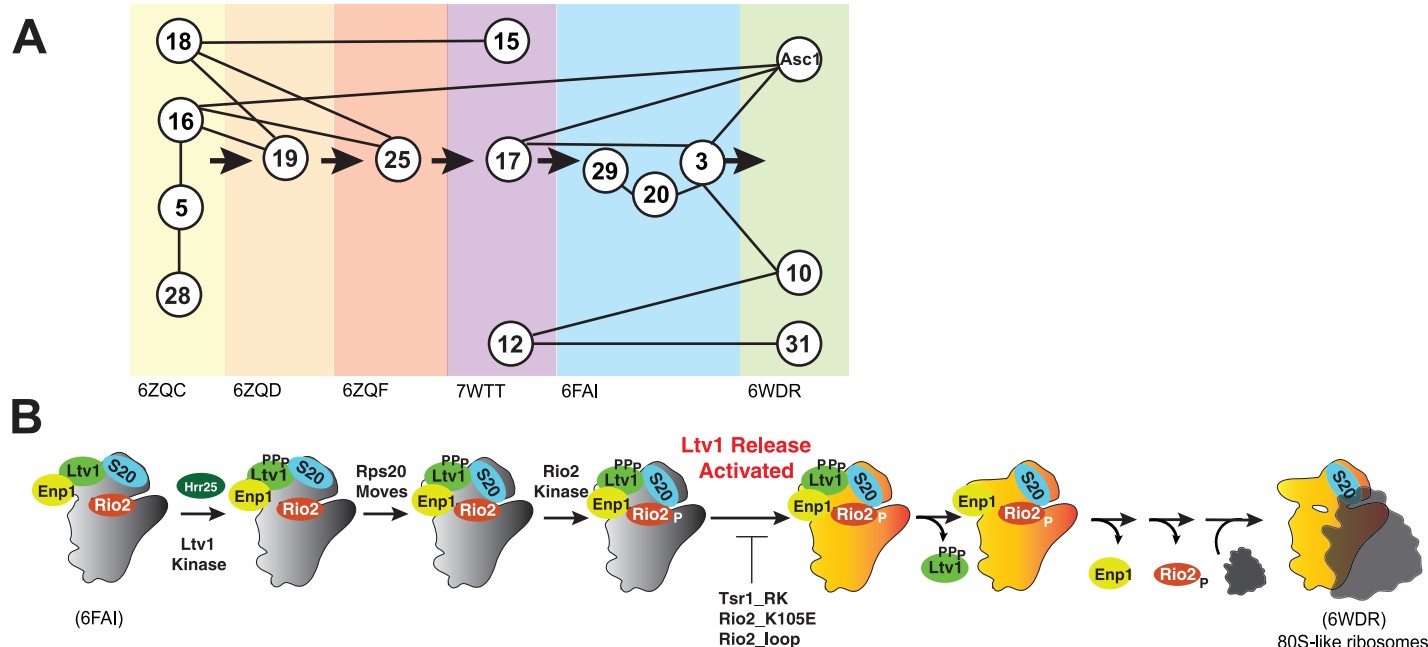

**Fig 1. Assembly of the small subunit head is hierarchical.** (A) Summary of the assembly hierarchy of ribosomal proteins to the head of the small ribosomal subunit. Assembly order was garnered from cryo-EM structures (indicated with the PDB ID at the bottom). Each intermediate is highlighted with a different color. Ribosomal proteins are indicated with the eukaryotic nomenclature. Contacts between different ribosomal proteins are indicated with line connections between the proteins, and arrows denote assembly progress. (B) A hierarchical set of steps regulates the formation of 80S-like ribosomes. Adapted from [21]. This cascade is initiated from stable pre-40S intermediate via phosphorylation of Ltv1 by the kinase Hrr25. While phosphorylation is necessary for Ltv1 release, it is not sufficient, and temporally separated. An unknown step that activates Ltv1 release was previously identified via its sensitivity to the Tsr1_RK (Tsr1_R709E, K712E), Rio2_K105E and Rio2_loop mutation [21].

recruitment of Rps25. However, other parts of the hierarchy are not explained in this manner: *e.g.*, it appears that the recruitment of Rps31 might be delayed, as it occurs much later than the binding of its only binding partner, Rps12, and the folding of its binding site (Fig 1A). Thus, the molecular forces underpinning the hierarchical assembly of RPs remain incompletely characterized.

The assembly factor Ltv1 has emerged as a critical player in 40S head assembly [20,21,24,26,31,32]. Binding directly to the universally conserved Rps3/uS3, Rps15/uS19, and Rps20/uS10 [24,32–34], it also plays roles in assembly of the eukaryote-specific Rps10 and Asc1 and was implicated in the folding of a critical 3-helix rRNA junction, j34-35-38 [14,15,22]. Moreover, it is a key actor in quality control of head assembly, which follows a hierarchical set of events that include the phosphorylation-mediated dissociation of Ltv1 (Fig 1B, [21]). Notably, even though Ltv1 phosphorylation is necessary for its dissociation, it is temporally separated, and first followed by rearrangements of Rps20/uS10, which in turn enable Rio2 phosphorylation. Rio2 phosphorylation then allows for an undefined step that activates Ltv1 release. Ltv1 release leads to folding of j34-35-38, dissociation of Enp1 and then Rio2, and ultimately the formation of 80S-like ribosomes, critical quality control intermediates that couple the final maturation steps to quality control. Importantly, this hierarchical sequence of events depends on the correct binding and positioning of Rps3/uS3, Rps15/uS19, and Rps20/uS10 as well as the correct folding of j34-35-38, and ensures that the newly-made subunits can accurately discriminate against non-AUG start codons [21]. Nevertheless, the full range of Ltv1 functions as well as their mechanistic basis has remained unclear as all studies have been carried out with Ltv1 deletion mutants, which tend to obscure later phenotypes.

Using a mutation in Ltv1 that causes a rare dermatological condition, LIPHAK syndrome [35], as well as a collection of genetic, biochemical, structural and mass spectrometry analyses, we demonstrate that Ltv1 plays global roles in assembly of the small subunit head, as nearly all RPs are depleted from this structure in ribosomes from cells with the LIPHAK mutation. DMS-MaP Seq was used to identify regions of change, and this analysis, combined with genetic and biochemical analyses demonstrates how Ltv1 establishes the hierarchy in RP binding, by delaying critical contacts between Rps3/uS3 and Rps20/uS10 to enable the recruitment of Rps29/uS14, and by regulating the recruitment of Rps31. Thus, our mechanistic analysis explains how incorporation of Rps29/uS14 and Rps31 is managed and explain why these are the two proteins most depleted from the subunit in cells with the disease-associated Ltv1_L216S. Thus, Ltv1 is a master regulator of assembly of the small ribosomal subunit head, with roles both in assembly and quality control.

## Results

### The disease-associated Ltv1_L216S mutation confers a growth phenotype in yeast

Recently, the same mutation in human Ltv1, Ltv1_N168S was recovered in two unrelated families suffering from LIPHAK syndrome, a rare dermatological disorder characterized by altered skin pigmentation arising from subcutaneous oxygenation defects [35]. We reasoned that this mutation could be used to uncover functions for Ltv1 in 40S ribosome assembly beyond those identified from the absence of the protein, which leads to an rRNA folding defect [21,22,26] that might mask additional deficiencies. We therefore set out to use our established yeast system to study defects in assembly that result from this mutation. Sequence alignments show that N168 in humans assumes the same position as L216 in yeast (S1A Fig). We therefore used our previously described Ltv1 deletion strain and plasmids encoding either wild type (wt) Ltv1 or Ltv1_L216S under the constitutive TEF promoter to test whether Ltv1_L216S confers a growth defect in yeast. Importantly, plasmid-driven expression of wt Ltv1 fully rescues the growth defect from the absence of Ltv1 (S1B Fig), validating the use of this system. Quantitative growth measurements show that indeed Ltv1_L216S produces a small growth defect (Fig 2A), which is specific for Ltv1_L216S and not observed for the neighboring Ltv1_L217S (S1C Fig). The magnitude of the growth defect is the same when Ltv1 is produced from the strong TEF promoter, and the weaker Cyc1 promoter, indicating that the defect does not arise from weak Ltv1 binding or low levels of the mutant Ltv1 protein (S1D Fig), consistent with Western blot analyses that show the same levels of wild type and mutant Ltv1 (S1E Fig). Thus, the disease-associated Ltv1_L216S mutation (LIPHAK mutation) causes a growth defect in yeast.

### Ltv1_L216S is mispositioned blocking its phosphorylation-dependent release

Ltv1 is a late-binding and acting 40S ribosome assembly factor. Thus, we wanted to confirm that the growth defect from the Ltv1_L216S mutation reflected a ribosome assembly defect, as expected. Absence of Ltv1 leads to defects in rRNA folding and recruitment of late-binding proteins Rps10 and Asc1 and reduces 40S ribosome levels [21,22,26]. Mutations that block its release are dominant negative [20], and impair the formation of so called-80S like ribosomes. 80S-like ribosomes are critical intermediates in 40S maturation that link maturation to quality control in tests that mimic ribosome functionality ([36,37], Fig 1B).

To test if the mutation affected the formation of 80S-like ribosomes, we used a previously described assay that takes advantage of the observation that 80S-like ribosomes accumulate to

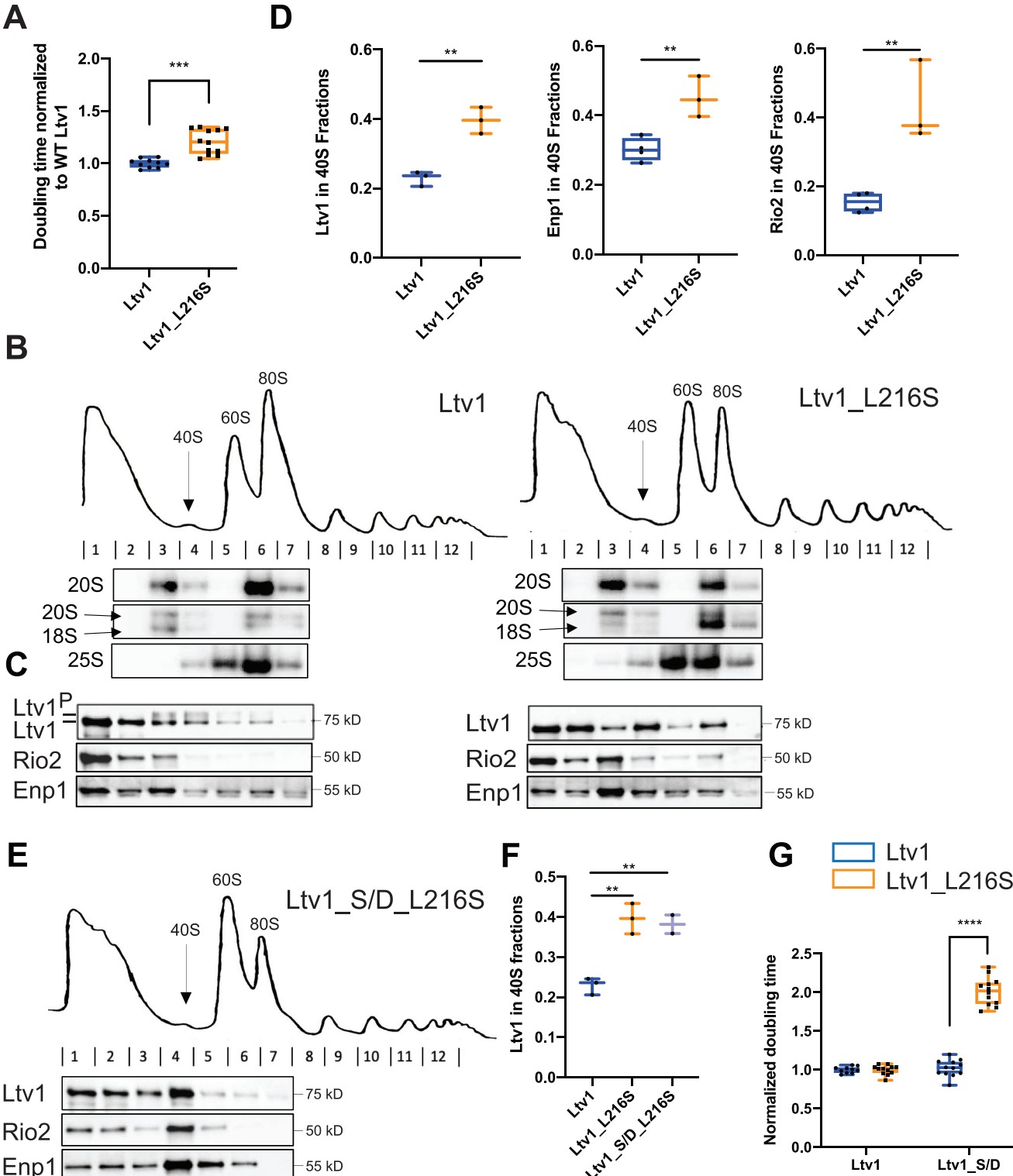

**Fig 2. The LIPHAK-syndrome associated Ltv1_L216S mutation affects 40S head assembly, by impairing Ltv1 phosphorylation.** (A) The Ltv1_L216S mutation produces a small growth defect. Growth of yeast strains lacking endogenous Ltv1 and expressing plasmid-encoded Ltv1 from the TEF promoter was measured, and doubling times extracted. Data are shown as box-and-whisker plots. Significance was tested using an unpaired t-test. ***, P<0.001. (B) Absorbance profiles at 260 nm of 10–50% sucrose gradients from Fap7 depleted cells expressing only plasmid-encoded wt Ltv1 or Ltv1_L216S. Northern blots of each gradient fraction are shown. (C) Western blots for Ltv1, Enp1 and Rio2 are shown. The position of phosphorylated Ltv1 is indicated with

Ltv1[P]. (D) Quantification of the data in panel C and biological and technical replicates, to demonstrate accumulation of Ltv1 (left), Enp1 (middle) and Rio2 (right) in pre-40S ribosomes. Data are shown as box-and-whisker plots. When fewer than 4 data points were obtained only the median and minimum/maximum, not the quartile are calculated. Significance was tested using an unpaired t-test. **, P<0.01; ***, P<0.001 (E) Absorbance profiles at 260 nm of 10–50% sucrose gradients from Fap7 depleted cells expressing only plasmid-encoded wt Ltv1 or Ltv1_S/D_L216S. Western blots for Ltv1, Enp1 and Rio2 are shown. (F) Quantification of the data in panel E and biological and technical replicates, to demonstrate accumulation of Ltv1 in pre-40S ribosomes. Significance was tested using an unpaired t-test. **, P<0.01. Note that data for WT Ltv1 and Ltv1_L216S are the same as in panel D and replotted for comparison. (G) Normalized (to WT Ltv1) doubling times from yeast expressing wt Ltv1, Ltv1_L216S, Ltv1_S/D or Ltv1_S/D_L216S demonstrate that the phosphomimetic Ltv1_S/D mutation [20] does not rescue, but instead exacerbates the effect from Ltv1_L216S. Significance was tested using a two-way ANOVA. ****, P<0.0001.

high levels in yeast strains depleted of the essential ATPase Fap7 [20,21,36]. Combining Fap7 depletion with mutations that impair the formation of 80S-like ribosomes reduces their accumulation. The formation of 80S-like ribosomes is then measured using sucrose gradient centrifugation, combined with Northern and Western analysis to identify the sedimentation of the 18S rRNA precursor (20S rRNA), and the remaining assembly factors, including Ltv1, respectively. We therefore analyzed lysates from yeast depleted of Fap7 and expressing wt or mutant Ltv1 using sucrose gradients and probed the fractions for 20S rRNA and assembly factors (Fig 2B and 2C). As shown in Fig 2B, Ltv1_L216S indeed shifted some of the 20S pre-rRNA from the 80S fraction back to the 40S fraction, demonstrating that the mutation impairs the formation of 80S-like ribosomes.

We and others have previously shown that the formation of 80S-like ribosomes requires the phosphorylation of Ltv1 [20,32], as well as the ordered release of Ltv1, Enp1 and Rio2 (Fig 1B, [21]). Intriguingly, the L216S mutation impairs Ltv1 phosphorylation (Fig 2C). Because Ltv1 phosphorylation is required for its release [20,32], Ltv1_L216S also accumulates in pre-40S ribosomes, with less of it in the free fraction (Fig 2C and 2D). Moreover, as Ltv1 release is required for Enp1 and Rio2 dissociation from pre-40S [21], the Ltv1 mutation also leads to accumulation of Enp1 and Rio2 in pre-40S (Fig 2C and 2D). Thus, together these data indicate that the Ltv1_L216S mutation impairs 40S ribosome maturation, by interfering with the phosphorylation of Ltv1, and its subsequent release, ultimately preventing the formation of 80S-like intermediates.

Phosphomimetic mutations in the Ltv1 phosphorylation site (S336D, S339D, S342D, Ltv1_S/D) can rescue the depletion of the Hrr25 kinase [20], and promote release of Ltv1 in the absence of Hrr25, as well as in mutants that block Ltv1 phosphorylation [21]. We therefore combined the phosphomimetic S/D mutation with L216S to produce Ltv1_L216_S/D and asked if the S/D mutation could rescue Ltv1 release as expected and as seen in all other cases. Surprisingly, Ltv1_L216_S/D remains accumulated in pre-40S ribosomes, and similarly, neither Enp1 nor Rio2 release was rescued (Fig 2E and 2F). Consistently, the S/D mutation exacerbated rather than rescued the growth defect from the Ltv1_L216S mutation (Fig 2G).

The observation that Ltv1_L216S cannot be phosphorylated by Hrr25, and that a phosphomimetic mutation cannot rescue the release or growth defects from the L216S mutation, suggest that Ltv1_L216S is positioned differently on pre-40S ribosomes, so that it is no longer accessible to the Hrr25 kinase, and such that S336, S339 and S342 no longer weaken binding when mutated to phosphomimetic aspartate residues. In contrast, the interface with Enp1 appears to remain largely unchanged: an Enp1 mutation, Enp1_WKK (W224V, K228E, K231E), rescues the non-phosphorylatable alanine mutations in Ltv1, Ltv1_S/A (S2A Fig), because it leads to weaker Ltv1 binding (S2B Fig). Enp1_WKK also rescues Ltv1_L216S (S2C Fig), suggesting that the interface between Enp1 and Ltv1 is the same in the Ltv1_S/A and Ltv1_L216S mutants.

Together, the data in this section demonstrate that Ltv1_L216S impairs ribosome assembly, because it cannot be phosphorylated and released, and strongly suggest that these defects arise because Ltv1 is mispositioned on pre-40S ribosomes. We also note that the small growth

defects of the Ltv1_L216S mutation, as well as most of the other mutations within this work indicate that these mutations do not block the individual steps that they are associated with but rather make them less likely to occur, thereby impairing assembly.

## Ltv1 wraps around the side of the head

To better understand how the Ltv1_L216S mutation alters its function in the context of the genetic and biochemical data that indicate mispositioning of the mutant protein, we wanted to visualize the location of the mutation in the context of the pre-40S structure. Unfortunately, only a small portion of Ltv1 has been visualized in current cryo-EM structures [14,15]. We therefore obtained a predicted structure for Ltv1 from the AlphaFold website (https://alphafold.ebi.ac.uk/), and then utilized the matchmaker function in Chimera [38] to place this structure on the pre-40S structure using the part of Ltv1 that has been previously visualized as a template.

Overall, this structure for Ltv1 and its placement in pre-40S does not lead to major steric clashes but is rather compatible with the pre-40S structure (Fig 3A). The composite structure predicts that Ltv1 contacts Rps3, Rps15, and Rps20, which are all validated binding partners for Ltv1 [24,32–34], thus supporting the placement. Moreover, aspartate 113 and glutamate 115 in Rps20, whose mutation to lysine (Rps20_DE) weakens Ltv1 binding to pre-40S [26], are at the predicted Ltv1/Rps20 interface (Fig 3B). Moreover, while Rps20_DE bind directly to K7/K10 in Rps3 (Rps3_KK), their mutation has opposite genetic interactions with mutations in or deletion of Ltv1, suggesting that this interaction was not made when Ltv1 was bound [26]. Indeed, the proposed structure places Ltv1 between those two RPs. Thus, the composite structure is consistent with and explains available data, suggesting that it is correct at least in broad strokes, although likely not in atomic detail.

To further validate this structure and the placement of Ltv1, we next designed point mutants in ribosomal proteins, which are expected to disrupt Ltv1/RP interactions, and then tested their genetic interaction with Ltv1_L216S.

The placement of the Ltv1 structure on the nascent 40S predicted an interaction between Ltv1 and residues K75 and R76 in Rps3 (Fig 3C). We therefore tested whether mutation of these residues to glutamates (Rps3_KR) influences cell growth. In fact, Rps3_KR grows over 3-fold more slowly than wt Rps3 (Fig 3D), demonstrating the importance of this residue for ribosome assembly and/or function. Next, we combined Rps3_KR with Ltv1_L216S to test for genetic interactions. Notably, Ltv1_L216S and Rps3_KR are epistatic, consistent with the idea that Ltv1 mispositioning in the Ltv1_L216S mutant perturbs the interaction between Ltv1 and Rps3_KR, therefore validating the position of Ltv1 in the composite structure. This observation also suggests that at least part of the growth defect in the Rps3_KR mutant reflects a defect in ribosome assembly.

In addition to interactions with Rps3, Rps15 and Rps20, which have been previously validated [24,32–34], the proposed structure of Ltv1 on pre-40S also predicts an interaction between Ltv1 and Rps12 (Fig 3E). To confirm this prediction and further validate the Ltv1 placement, we tested for genetic interactions between Rps12 and Ltv1. The predicted point of contact between these two proteins includes Rps12_R108. We therefore mutated this residue to alanine (Rps12_R108A). Indeed, when combined with Ltv1_L216S, Rps12_R108A is epistatic (Fig 3F), supporting an interaction between the two proteins. Synthetic negative interactions between Rps12 deletion and Ltv1 deletion or Enp1 point mutants have also been previously reported [39] and are consistent with an interaction between these proteins.

Together, these mutational and genetic analyses, combined with the previous biochemical data validate the structure of Ltv1. Additional validation comes from the analysis of rRNA mutations in the next section.

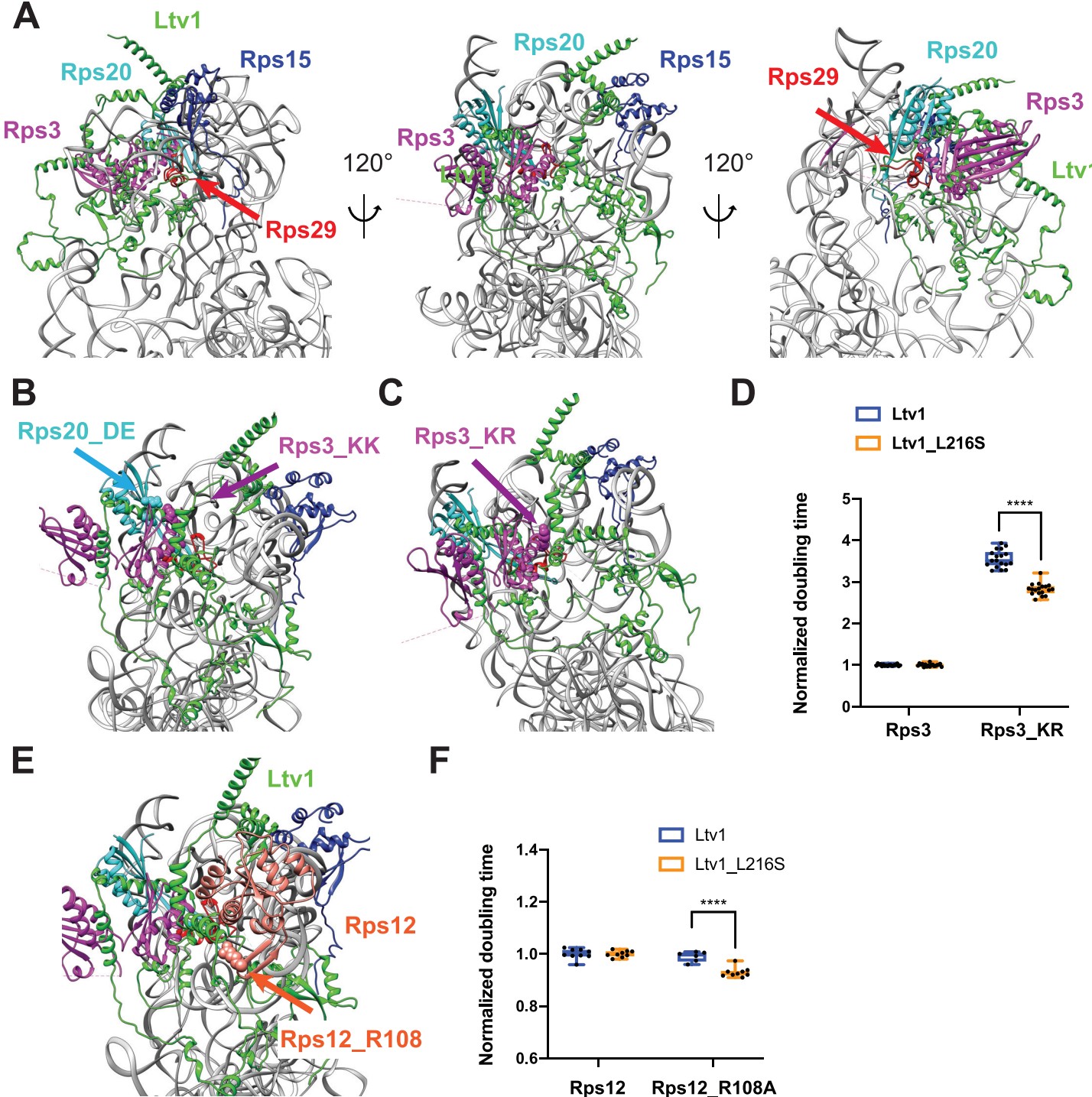

**Fig 3. Ltv1 binding to pre-40S ribosomes.** (A) A structure for Ltv1 was obtained from the AlphaFold website (https://alphafold.ebi.ac.uk/) and placed on the structure for pre-40S ribosomes (PDB ID 6FAI), by overlay with the Enp1-binding alpha-helix. The composite structure does not produce significant clashes with pre-40S ribosomes and shows interactions between Ltv1 and Rps3, Rps15 and Rps20. All structure displays were generated with Chimera [38]. (B) Detail of the complex predicts that Ltv1 lies between Rps20_DE (D113,E115) and Rps3_KK (K7, K10, highlighted in cyan and purple spheres, respectively), consistent with previous biochemical and genetic data. (C) Detail of the complex predicting an interaction between Ltv1 and residues K75 and R76 in Rps3 (Rps3_KR, highlighted in purple spheres). (D) Doubling times (normalized to wt Rps3) for yeast cells expressing either wt Rps3 or Rps3_KR and either wt Ltv1 or Ltv1_L216S. Significance was tested using a two-way ANOVA. ****, P<0.0001. (E) Detail of the complex predicting that Ltv1 binds Rps12, with a contact from Rps12_R108 (highlighted in red sphere). (F) Doubling times (normalized to wt Rps12) for yeast cells expressing either wt Rps12 or Rps12_R108A and either wt Ltv1 or Ltv1_L216S. Significance was tested using a two-way ANOVA. ****, P<0.0001.

## Ltv1_L216S is located adjacent to helix 34 and perturbs folding of junction j34-35-38

The structure predicts that L216 is part of a short alpha-helical segment adjacent to helix 34 (h34, Fig 4A). H34 is part of a 3-helix junction with helices 35 and 38 (j34-35-38). In mature 40S, j34-35-38 forms a tertiary interaction with the helical loop closing helix 31 (l31). We have previously shown that j34-35-38 is misfolded in the absence of Ltv1 [21,22,26]. Moreover, our biochemical and previous structural data indicate that j34-35-38 folds once Ltv1 dissociates from pre-40S ribosomes [14,15,22]. These observations are consistent with the position of Ltv1 adjacent to h34 predicted by the Ltv1-pre-40S structure here and indicate a direct role for Ltv1 in chaperoning the folding of this rRNA segment.

However, the structure also Ltv1 contacts l31 (Fig 4B). To validate this suggested interaction and test for its importance, we mutated residues Y82, D83, and Y84 to make Ltv1_YDY (Ltv1_Y82V; D83A; Y84V). This mutation produces a small growth defect (Fig 4C), therefore demonstrating the importance of this predicted interaction and further validating the structure. Moreover, this interaction between Ltv1 and l31 opened the possibility that the absence of Ltv1 leads to j34-35-38 misfolding via j34-35-38 directly, or via l31.

To test whether Ltv1 chaperones the folding of j34-35-38 directly, or rather affects its folding indirectly by preventing the premature folding of l31, or both, we carried out a series of genetic experiments. Deletion of snR35, a snoRNA which directs pseudouridylation of U1191 in l31, and prevents the premature folding of l31, results in misfolding of j34-35-38 [22]. This is because the folding of l31 stabilizes misfolded structures in j34-35-38. If a mutation in Ltv1 is epistatic with snR35 deletion, it would suggest that it leads to premature folding of l31 in the same way as snR35 deletion does. In contrast, synergistic effects would indicate that its effects are not via l31. We therefore combined the Ltv1_L216S mutation with the snR35 deletion. The data in Fig 3D show that deletion of snR35 is synthetically sick with the Ltv1_L216S mutation, indicating that the effects on j34 misfolding arise directly from its location at j34. Nonetheless, it is difficult to rule out that additional effects might be due to premature folding of l31.

To further confirm this conclusion, we next tested whether rRNA mutations in the helices that surround the junction and are adjacent to Ltv1 would be sensitive to the Ltv1_L216S mutation. We therefore mutated U1285, U1286 and A1422, which are part of a bulge at the top of helix 34, as well as G1288 and C1327, the base pair which closes h35 (Fig 4E), and tested their growth defects alone, and in combination with Ltv1_L216S.

For this purpose, we deleted the Ltv1 gene from a strain where RNA polymerase I has a mutation that renders it temperature sensitive, thereby shutting off all transcription from the endogenous rDNA genes above 32°C [40]. When supplemented with plasmids encoding either wt rRNA, or a mutation of interest, their impact can then be tested. In addition, we have also supplied either wt Ltv1 or Ltv1_L216S. By themselves the rRNA mutations have moderate to no effect on growth (S3 Fig). However, when combined with Ltv1_L216S, they demonstrate synthetic sick effects (Fig 4F). Together, these data demonstrate a direct role for Ltv1 in enabling the correct folding of j34. Moreover, the observation of j34-35-38 misfolding in the absence of or with Ltv1 mutants further validates the structure, which places Ltv1 directly adjacent to j34-35-38.

## Head protein assembly is globally perturbed in the Ltv1_L216S mutant

Given that the Ltv1_L216S mutant is only partially functional, we wanted to use its defects to gain additional insight into the functions of the wild type protein. Our previous work had shown that Asc1 and Rps10 were substoichiometric in ribosomes from cells lacking Ltv1 [26], suggesting mispositioning of Rps3, their common binding partner. Moreover, genetic

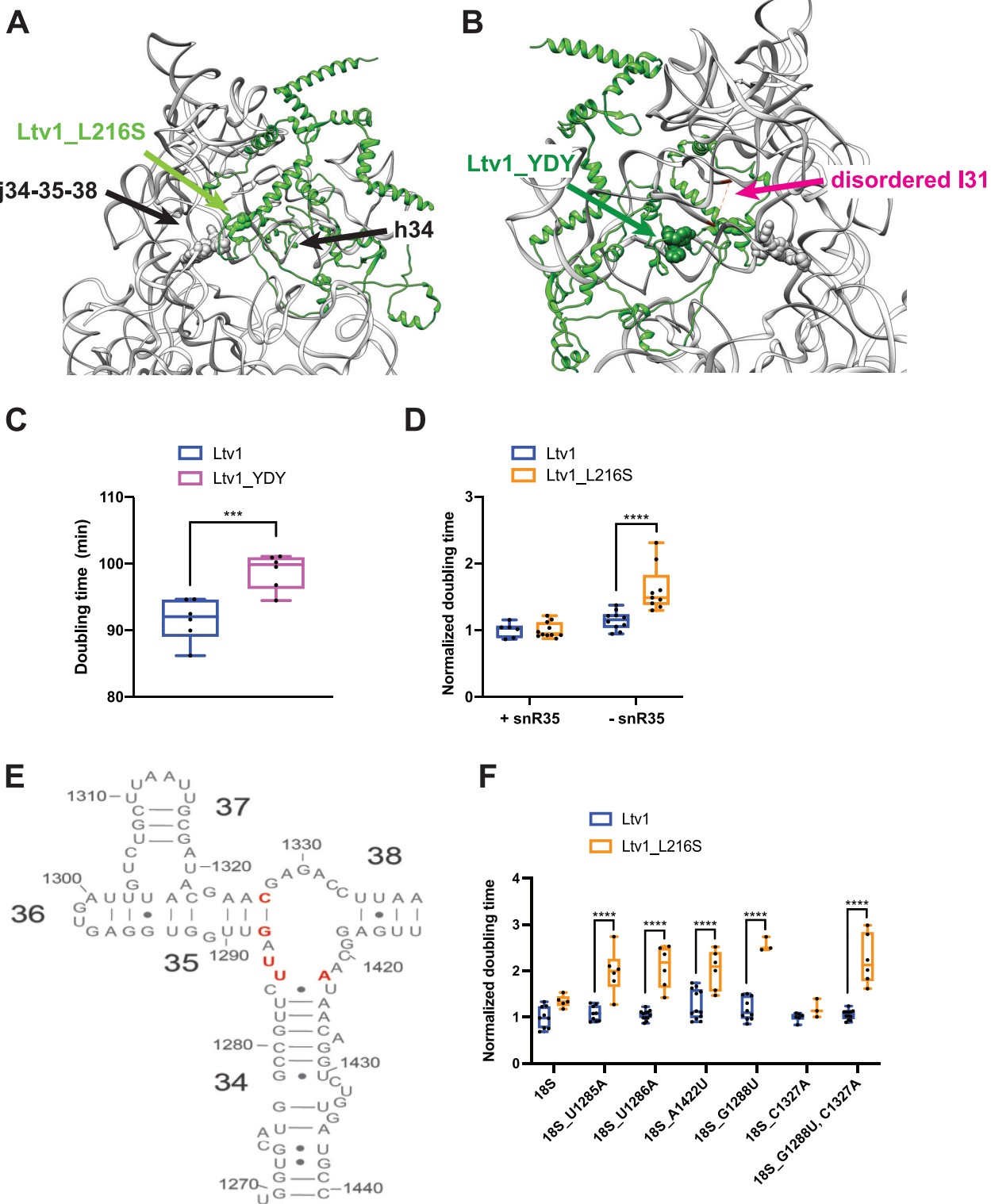

**Fig 4. The LIPHAK syndrome associated mutation Ltv1_L216S binds adjacent to j34-35-38 and affects its folding.** (A) Detail of the complex demonstrating that Ltv1_L216 (highlighted in green spheres) binds adjacent to j34-35-38 and h34. Residues mutated in panel E are highlighted in gray sphere. (B) Detail of the complex demonstrating that Ltv1_YDY (Y82A,D83R,Y84A, highlighted in dark green spheres) binds adjacent to l31 (which is disordered in this structure and thus not visible). Its start and end are indicated in magenta and connected by a dashed line. Residues mutated in panel E are highlighted in gray sphere. (C) Doubling for yeast cells expressing either wt Ltv1 or Ltv1_YDY. Significance was tested using

an unpaired t-test. ***, P<0.001. (D) Doubling times (normalized to wt snR35) for yeast cells containing or lacking snR35 and expressing either wt Ltv1 or Ltv1_L216S. Significance was tested using a two-way ANOVA. ****, P<0.0001. (E) Secondary structure diagram of j34-35-38 illustrating the locations of the mutations in panel F. (F) Doubling times (normalized to wt Ltv1) for yeast cells containing a temperature sensitive RNAPol I mutation (NOY504) and expressing plasmid encoded wt or mutant 18S rRNA and wt Ltv1 or Ltv1_L216S. Significance was tested using a two-way ANOVA. ****, P<0.0001.

experiments also indicated Rps20 was mispositioned or reduced. To investigate RP recruitment to ribosomes globally, we purified 40S ribosomes from yeast expressing either wt Ltv1 or Ltv1_L216S, and then analyzed RP composition via semi-quantitative mass spectrometry. We obtained three independent biological replicates from each strain, normalized the peptide reads for each protein to the reads for the entire sample to normalize for differences in sample abundance, and quantified RP occupancy in the L216S mutant relative to wt Ltv1. These data reveal a striking depletion of most RPs, except Rps3, Asc1, Rps20 and Rps28, from the small ribosomal subunit head in the ribosomes from mutated yeast (Fig 5A and 5B). In contrast, RPs from the body or platform were largely unaffected.

## DMS-MaPseq reveals global roles for Ltv1 in 40S structure

To better understand the molecular basis for the loss of many head RPs and determine whether additional rRNA regions are misfolded in the ribosomes from the L216S mutant yeast, we utilized DMS-MaPseq to probe the structure of mature 40S subunits from yeast expressing wild type Ltv1 or L216S Ltv1. As previously described [22, 41], ribosomes were exposed to DMS or mock treatment, and after quenching the reaction, 18S rRNA was fragmented, reverse transcribed, and sequencing libraries prepared. After sequence alignment, modified nucleotides are misread in the reverse transcription reaction and appear as a "mutation" relative to the genomic DNA (S4A Fig). Thus, the mutational rate at each nucleotide is a measure of its propensity to be modified by DMS. Analysis of the sequencing data indicates high read density over the entire molecule (S4B Fig), with some drop off near the 3'-end, due to the methylation of A1781/A1782 (S4C Fig). We next compared the mutational rate for each nucleotide in wt and mutant cells, identifying 14 residues, that were more accessible in the mutant and 1 residue that was more protected in the mutant (Fig 6A and 6B and S1 Table).

Six of the 18S rRNA residues with altered DMS accessibility are located in the 40S head. In addition, there is an additional cluster of 5 residues with altered DMS accessibility on the subunit interface (Fig 6B). This cluster will be discussed separately below (*The Ltv1_L216S mutation impairs movement of Tsr1 to the beak*).

## The L216S mutation leads to premature formation of the Rps20-Rps3 contact, which blocks Rps29 binding

Four of the residues that are more DMS accessible in ribosomes from Ltv1_L216S cells, A1189, A1196, C1197 and A1515 are directly adjacent to residues in Ltv1 (Fig 6C), further supporting the suggested placement of Ltv1 in pre-40S ribosomes. However, none of them are close to L216, strongly supporting the conclusion above, based on genetic interactions, that Ltv1 is globally mispositioned in the Ltv1_L216S mutant.

Of the residues with altered DMS accessibility, A1515 is directly adjacent to the residues mutated in Rps20_DE and Rps3_KK, which form a contact in mature 40S subunits. Previous data had suggested that Ltv1 blocks the formation of this contact between Rps20 and Rps3 [26], as they have opposite genetic interactions, and because mutation of Rps20, but not Rps3 weakens Ltv1 binding. Consistent with these previous data, the structure of Ltv1 on pre-40S

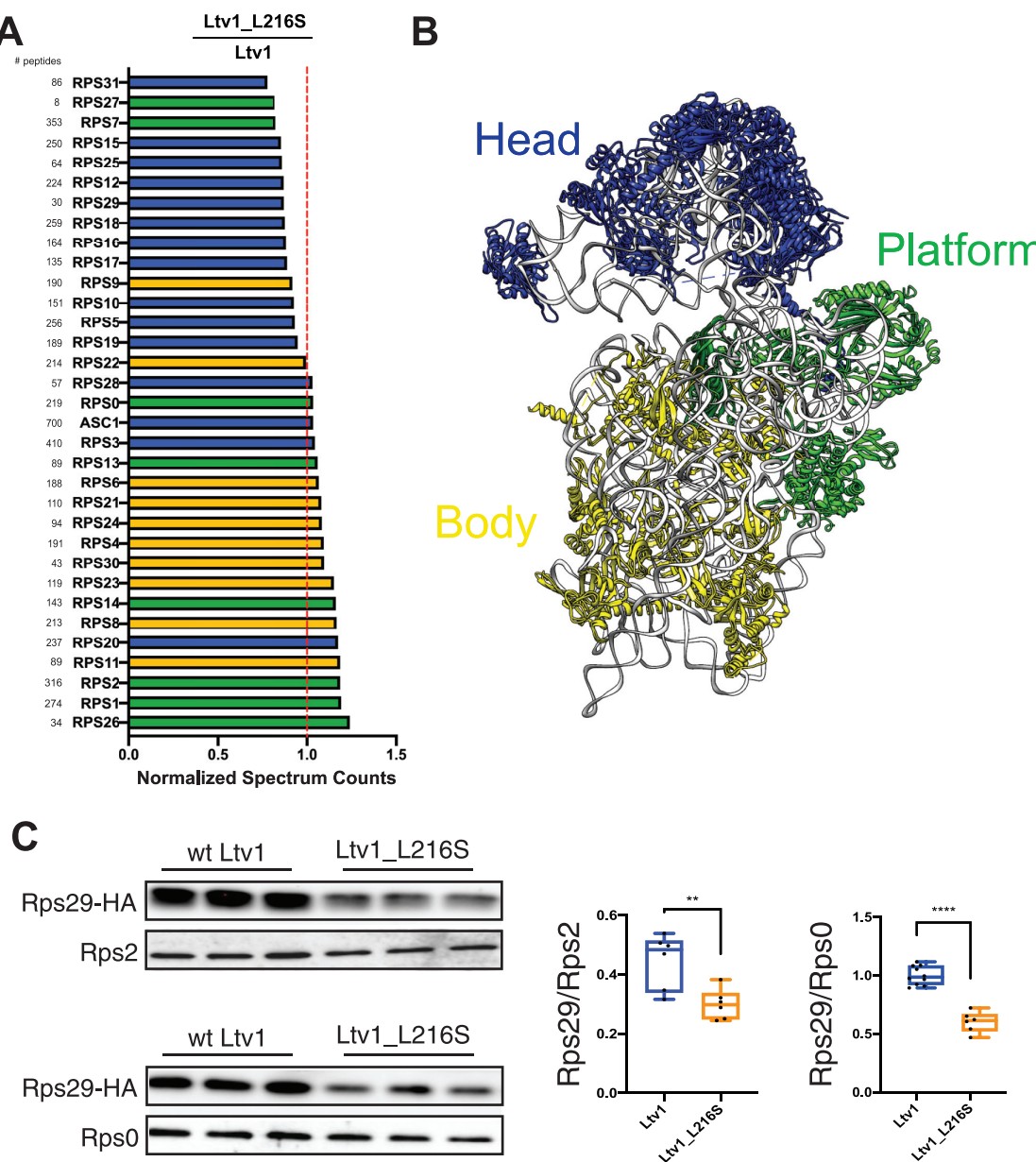

**Fig 5. The Ltv1_L216S mutation globally disrupts head assembly.** (A) Mature ribosomes from yeast expressing wt Ltv1 or Ltv1_L216S were purified and analyzed by mass spectrometry. Spectral counts for each protein in each sample were normalized by the total number of spectra, and abundance of each protein in Ltv1_L216S ribosomes normalized by the abundance in wt Ltv1 ribosomes. Number of spectra for each protein are indicated on the far left. Ribosomal proteins are color-coded for the substructure to which they bind, with head-binding proteins in blue, platform proteins in green, and proteins from the body in yellow. Averaged data from three biological replicates are shown. (B) Structure of mature 40S ribosomes highlighting the different substructures with the same color-code as in (A). (C) Western analysis of ribosomes purified from cells expressing Rps29-HA and either wt Ltv1 or Ltv1_L216S. Rps29 occupancy was quantified relative to Rps2 and Rps0. Significance was tested using an unpaired t-test. **, P<0.01; ****, P<0.0001.

predicts that Ltv1 is placed between these residues, interrupting this contact (Figs 6C and 3B). Thus, the increased accessibility of A1515 in the mutant ribosomes indicates that the L216S mutation leads to repositioning of Ltv1 near Rps3 and Rps20. Presumably this would allow for the premature formation of the contact between these two proteins. Consistently, Rps3 and

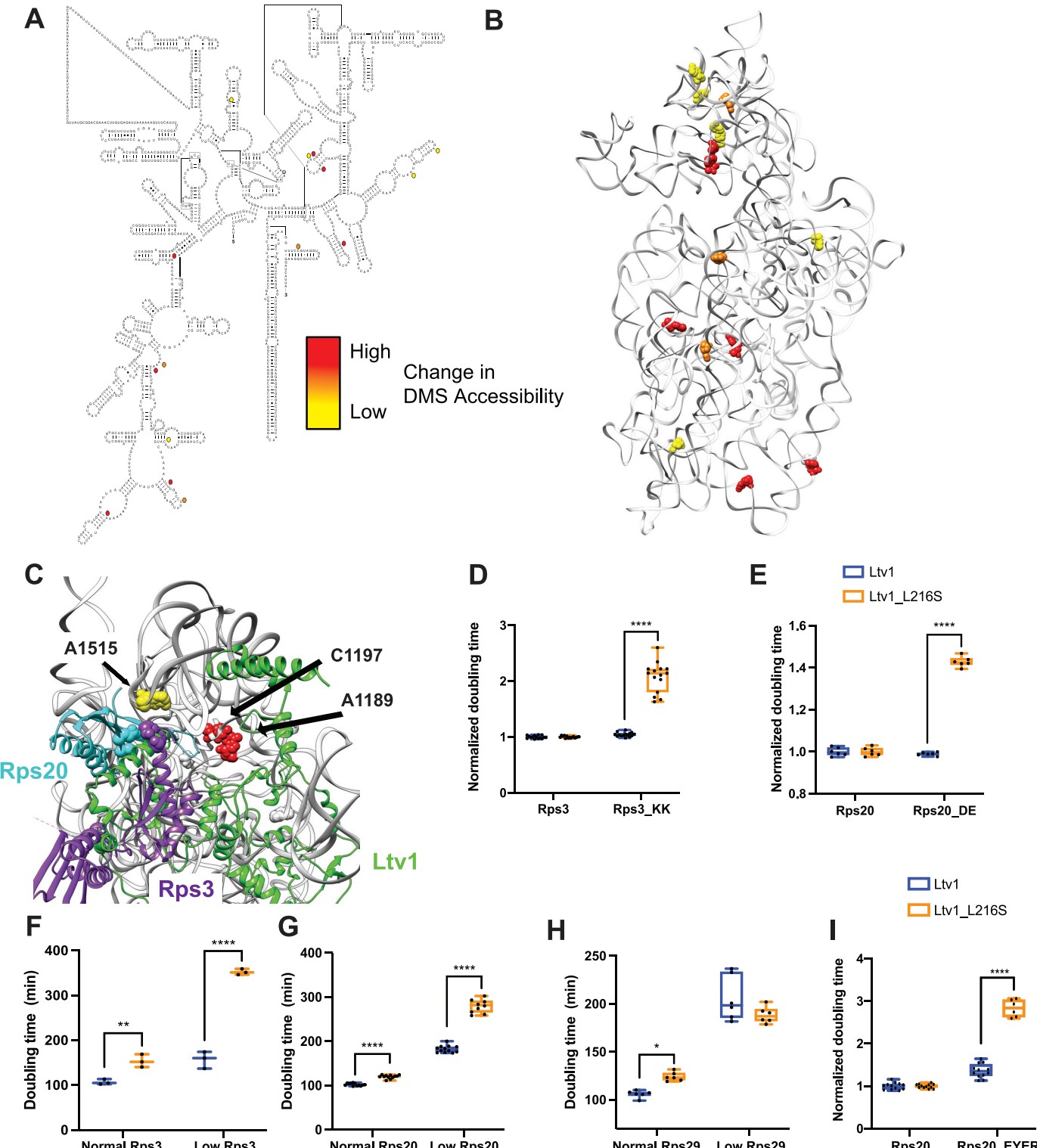

**Fig 6. DMS MaPSeq demonstrates perturbations in rRNA folding arising from the Ltv1_L216S mutation.** (A) Secondary structure of 18S rRNA demonstrating residues significantly altered in ribosomes from Ltv1_L216S cells. All residues (except C191) are more accessible in ribosomes from Ltv1_L216S. The extent of the alteration is indicated by yellow, orange or red circles. (B) Residues significantly altered in ribosomes from Ltv1_L216S cells mapped onto the 3D structure of 18S rRNA from pre-40S ribosomes (PDB ID 6FAI). Color scheme as in panel A. (C) Highlight of the composite structure from Fig 3, demonstrating that A1515, A1189 and C1197 are adjacent to residues in Ltv1. A1196 is not resolved in the structure and therefore not displayed. Rps20_DE (D1123,E115) and Rps3_KK (K7,K10) are shown in cyan and purple spheres, respectively. (D) Normalized doubling times for yeast cells expressing either wt Ltv1 or Ltv1_L216S and

either wt Rps3 or Rps3_KK (K7A,K10A). (E) Normalized doubling times for yeast cells expressing either wt Ltv1 or Ltv1_L216S and either wt Rps20 or Rps20_DE (D113R,E115R). (F) Doubling times for yeast cells expressing either normal or low levels of Rps3 and wt Ltv1 or Ltv1_L216S. Normal levels of Rps3 and Rps20 were obtained by using the TEF2 promoter-driven plasmids. Low levels of Rps3 and Rps20 were obtained by using Tet-promoter-driven plasmids and addition of 0 or 20 ng/ml of doxycycline (dox), for Rps3 and Rps20, respectively. (G) Doubling times for yeast cells expressing either normal or low levels of Rps20 and wt Ltv1 or Ltv1_L216S. Data are shown as box-and whisker plots. When fewer than 4 data points were obtained only the median and minimum/maximum, not the quartile are calculated. (H) Doubling times for yeast cells expressing either normal or low levels of Rps29 and wt Ltv1 or Ltv1_L216S. Low levels of Rps29 were obtained by using a Tet-promoter-driven plasmid, which even without dox addition produces reduced levels of Rps29. Normal levels of Rps29 were produced with TEF2 promoter-driven plasmids. (I) Doubling times for yeast cells expressing either wt Rps20 or Rps20_EYER (E80K,Y82A,E83K,R85E) and wt Ltv1 or Ltv1_L216S. Significance for panels D-I was tested using a two-way ANOVA. ****, P<0.0001.

Rps20 are (together with Asc1, which binds directly to Rps3), the only proteins that are *not* depleted from the head of the 40S subunit in ribosomes from Ltv1_L216S cells (Fig 5A and 5B). We therefore hypothesized that the premature formation of the Rps3-Rps20 contact would lead to mispositioning of Rps3/Rps20 and/or impair the recruitment of other RPs.

To test if the premature formation of the Rps3-Rps20 contact would lead to mispositioning of Rps3/Rps20, we combined the Rps20_DE and the Rps3_KK mutations with Ltv1_L216S. Indeed, both Rps3_KK and Rps20_DE demonstrate strong synthetic growth defects with Ltv1_L216S (Fig 6D and 6E). Moreover, reducing the expression of Rps3 or Rps20 using a tet-repressible promoter also is synthetically sick with the Ltv1_L216S mutation (Fig 6F and 6G). Thus, repositioning of Ltv1 due to the L216S mutation to allow for the premature formation of the contact between these two proteins impairs proper positioning of Rps3 and Rps20, rendering them sensitive to depletion and mutation.

To determine whether premature formation of the Rps3/Rps20 contact also impaired the recruitment of other RPs, we considered the structural data illuminating the folding of the head domain rRNA and the recruitment of its RPs [17]. Binding of Rps20 coincides with the appearance of helix 39 (h39) and a bulge in h41, which covers Rps29, in cryo-EM structures (S5A and S5B Fig), presumably because Rps20 stabilizes the tertiary contact between h39 and h41 in the head. Notably, these two structural elements obstruct the access of Rps29. Moreover, Rps29 is located under Rps20 (S5C and S5D Fig). Thus, this structural analysis predicts that Rps29 binds prior to Rps20, as Rps20 binding blocks Rps29 access both directly, and indirectly by promoting the folding of h39 and h41.

Indeed, the mass-spectrometry data indicated that Rps29 is one of the most-depleted proteins in the ribosomes from Ltv1_L216S cells (Fig 5A). However, Rps29 was detected by relatively few peptides, rendering this measurement unreliable. We therefore used Western analysis with HA-tagged Rps29 to confirm the mass spectrometry data. We constructed a yeast strain where Rps29 is HA-tagged and Ltv1 deleted, supplemented this strain with wt and mutant Ltv1 and purified ribosomes from this strain. Western blotting against the HA-tag on Rps29 demonstrates that indeed Rps29 is reduced by ~ 50% in ribosomes from Ltv1_L216S yeast (Fig 5C).

Next, we used genetic analyses to obtain additional data to support a model that in wild type cells Rps29 binds before Rps20, and that in the Ltv1_L216S mutant Rps29 is partially excluded from 40S subunits because of a change in the order of binding. We have previously described a mutant in Rps20, Rps20_EYER, in which four amino acids at the interface with Rps29 are mutated (E80K, Y82A, E83K and R95E, [21]). If Rps29 binds before Rps20, then this mutant should impair the recruitment of Rps20; in contrast, if Rps20 binds before Rps29, then this mutant should impair the recruitment of Rps29 (S5E Fig). We therefore devised a genetic strategy to distinguish between the two as described below.

We measured the effect from depletion of Rps29 on growth in wild type cells and yeast expressing Ltv1_L216S using yeast strains, where the Rps29 concentration can be reduced via

addition of doxycycline (dox), and which express either wt Ltv1 or Ltv1_L216S. While reducing Rps29 concentration reduces growth in yeast with wt Ltv1, it has no effect on the growth of yeast expressing Ltv1_L216S (Fig 6H). This surprising finding strongly suggests that in yeast expressing Ltv1_L216S Rps29 association is not rate-limited by Rps29 association, but rather a step that is independent of Rps29 concentration, such as a folding step, or the dissociation of Rps20.

Next, we used the observation that depletion of Rps29 is epistatic with Ltv1_L216S, while depletion of Rps20 is synthetically sick with Ltv1_L216S, to test if the Rps20_EYER mutation behaves akin to the Rps20 depletion or the Rps29 depletion. Indeed, when we combine Rps20_EYER with Ltv1_L216S, we observe synthetically sick interactions (Fig 6I), akin to the observation with Rps20 depletion (Fig 6G), and different to the effect from Rps29 depletion (Fig 6H). The observation that yeast with Rps29_EYER, which disrupts the Rps20/Rps29 interface, behave like Rps20 is depleted and not like Rps29 is depleted, suggests strongly that Rps29 is already present before this mutant binds, thus leaving Rps29 binding unaffected, while weakening the binding of Rps20 to the already prebound Rps29 (S5E Fig).

Strongly supporting these genetic interactions, are direct measurements of Rps20 occupancy in ribosomes from wt yeast or yeast expressing Rps20_EYER [26], which shows reduced levels of Rps20 in ribosomes from yeast expressing Rps20_EYER [26]. Moreover, Rps20_EYER impairs the phosphorylation of Ltv1, akin to mutations that misposition Rps20 [21]. Thus, the reduced binding of Rps20 to ribosomes in the Rps29_EYER mutant is not consistent with Rps20 binding prior to Rps29, and instead supports binding of Rps29 prior to Rps20, as predicted from structural considerations (S5 Fig). In addition, the Rps29 concentration independence in the Ltv1_L216S mutant suggests that in this mutant Rps29 binding requires a step that is Rps29-independent, such as dissociation of Rps20, or remodeling of rRNA structure. Thus, together these data strongly support a model whereby in wild type cells Rps29 binds before Rps20, and where this order is perturbed in the Ltv1_L216S mutant, leading to the requirement for a conformational step (rRNA remodeling or Rps20 dissociation), and reduced levels of Rps29 in ribosomes. We note that future experimentation might reveal additional roles for Ltv1 in the recruitment of Rps29, including some that take advantage of the predicted direct binding between Ltv1 and Rps29 (Fig 3).

## The Ltv1_L216S mutation affects an interaction between the C-terminal tails of Ltv1 and Rps15 with Tsr1

As described above, a second cluster of RNA residues altered in its DMS accessibility in the L216S mutant is on the subunit interface (Figs 6B, 7A and 7B). Intriguingly, four of the subunit interface rRNA residues altered in the Ltv1_L216S mutant are located around the experimentally determined binding site for the assembly factor Tsr1 (Fig 7A), in particular around an N-terminal helix, which acts as a hinge to allow it to swing towards the beak in 80S-like ribosomes [16], indicating that altered functionality of Tsr1 is responsible for their altered DMS accessibility. On the subunit interface Tsr1 is close to and likely interacts with the C-terminal tails (CTT) of Ltv1 and Rps15 (S6A Fig), which could both mediate an effect from the Ltv1_L216S mutation on Tsr1. Because the resolution of the available cryo-EM structures in that area is limited and does not unambiguously identify these three proteins, and because the AlphaFold-predicted structure shows a different path for yeast Ltv1, we first used genetic interactions to obtain evidence for interactions between Ltv1, Rps15 and Tsr1.

We have previously shown that the growth defect from a mutation in Tsr1, Tsr1_RK (R709E, K712E), which mutates an interaction between Tsr1 and rRNA in the head (Fig 7B, see S6A Fig for an illustration of the location for mutations in this section), is exacerbated by

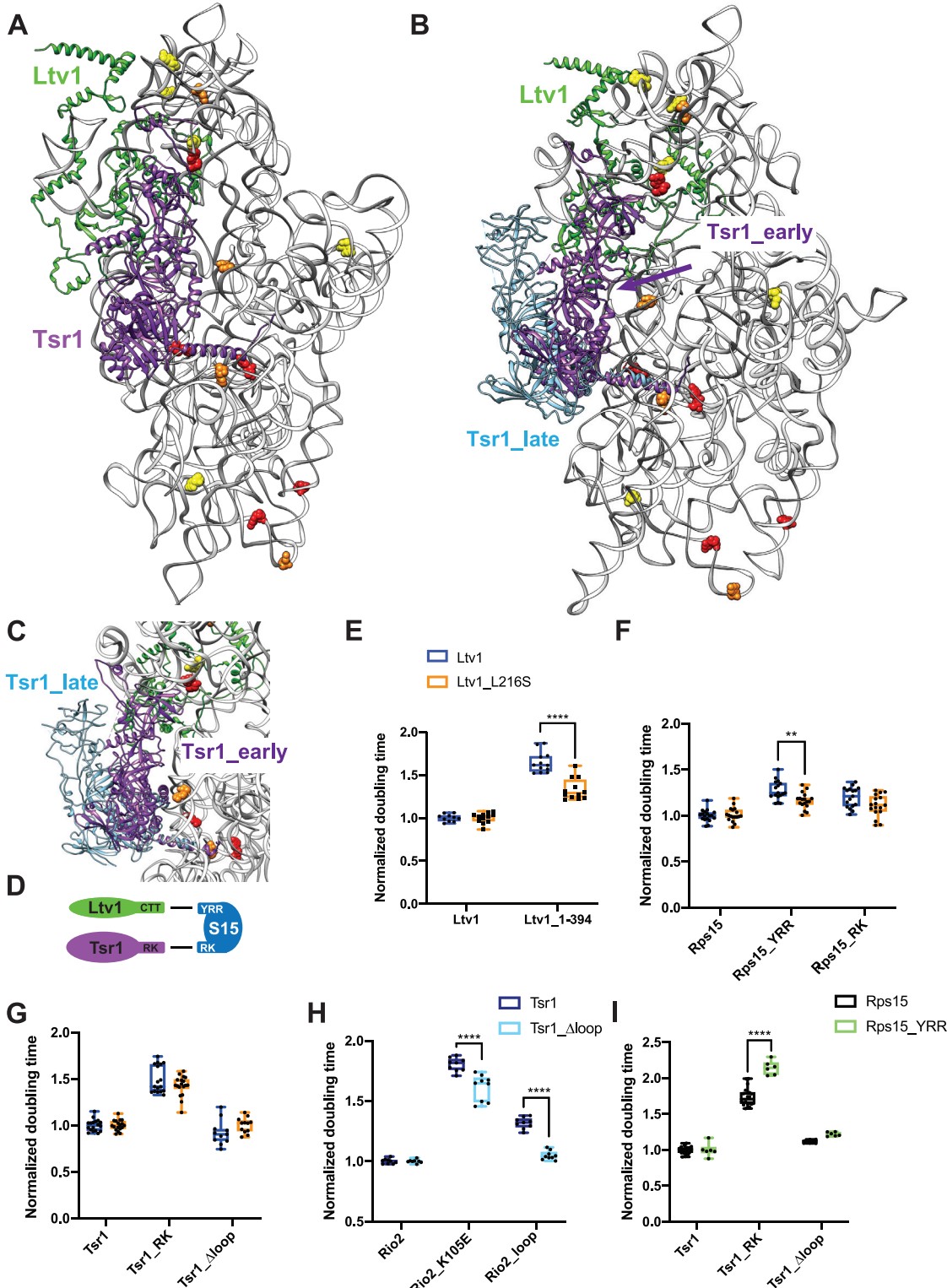

**Fig 7. Ltv1 dissociation requires Tsr1 movement to the beak.** (A) Structure of the pre-40S ribosome highlighting residues altered in ribosomes from Ltv1_L216S yeast as determined by DMS-MaPseq (Fig 6A and 6B) showing Tsr1 and Ltv1 (from PDB ID 6FAI). (B) Structure of the pre-40S ribosome with Ltv1 and Tsr1 in the position from earlier pre-40S (6FAI, purple) and from 80S-like ribosomes (in blue), near the beak (6WDR). 6FAI and 6WDR were overlaid by matching the 18S rRNA. (C) Structural detail of the complex in B, viewed from the top of the subunit, demonstrating that Tsr1_early, but not Tsr1_late blocks binding and dissociation

of Ltv1. (D) Summary of the genetic interaction network shown by the data in panel E and F and in S5 Fig. (E) Doubling times (normalized to wt Ltv1) for yeast cells expressing either wt Ltv1 or Ltv1_L216S in full length or truncated (at amino acid 394) form. (F) Doubling times (normalized to wt Rps15) for yeast cells expressing either wt Rps15, Rps15_YRR (Y123I,R127K,R130K) or Rps15_RK (R137E,K142E) and wt Ltv1 or Ltv1_L216S. (G) Doubling times (normalized to wt Tsr1) for yeast cells expressing either wt Tsr1 or Tsr1_RK or Tsr1Δloop and wt Ltv1 or Ltv1_L216S. (H) Doubling times (normalized to wt Rio2) for yeast cells expressing either wt Rio2, Rio2_K105E or Rio2_loop. Data for Tsr1_RK and Tsr1_Δhinge are in S5e Fig of [16] (I) Doubling times (normalized to wt Rio2) for yeast cells expressing either wt Rio2, or Rio2_K105E. Data for Tsr1_RK (R709E,K712E) and Tsr1_Δhinge are in S5g Fig of [16]. Significance for panels E-I was tested using a two-way ANOVA. **, P<0.01; ***, P<0.001; ****, P<0.0001.

deletion of the CTT of Ltv1 after amino acid 394, Ltv1_1–394 [16]. Moreover, Tsr1_RK is rescued by a cancer-associated mutation in the CTT of Rps15, Rps15_S136F [21]. These data link Tsr1, Ltv1 and Rps15, and are supported by synthetic negative genetic interactions between Tsr1_RK and Rps15_YRR (Y123I, R127K, R130K [16]). However, because negative genetic interactions are often less reliable and can arise from global perturbations in ribosome assembly, and we had only one rescue, we sought to strengthen the genetic interaction network between Tsr1 and the CTTs of Ltv1 and Rps15.

Notably, we show here that the previously described rescue of Tsr1_RK by an internal deletion in Tsr1, Tsr1_Δloop, requires the CTT of Ltv1 (S6B Fig). Moreover, Tsr1_Δloop also rescues the phosphorylation-deficient Ltv1_S/A (S6C Fig). Together, these data provide strong evidence for a functional connection between Tsr1 and Ltv1.

Tsr1_RK also demonstrates synthetic genetic interactions with Rps15_YRR, but epistatic interactions with Rps15_RK (S6D Fig). These mutations in Rps15 are all in the CTT. In contrast, Ltv1_1–394 is neutral to Rps15_YRR, but synthetically sick with Rps15_RK (S6E Fig). Together, these data strongly support a functional network between Tsr1 and the CTTs of Ltv1 and Rps15, consistent with the available cryo-EM structures. Moreover, the epistasis data also indicate that the Ltv1_CTT works through the Rps15_YRR residues to affect Rps15, while Tsr1 works through the Rps15_RK residues (Fig 7D), the latter being supported by the cryo-EM data (S6A Fig).

To decipher if this functional interaction between Ltv1, Rps15 and Tsr1 is disrupted in the Ltv1_L216S mutant, we first combined the Ltv1_L216S mutation with the truncation of its CTT, Ltv1_1–394, and measured the growth effects from these mutations quantitatively. Indeed, these two mutations are epistatic (Fig 7E), strongly suggesting that L216S disrupts the interaction between the CTTs of Ltv1 and Rps15. Moreover, the epistatic effect also suggests that the Ltv1 CTT is mispositioned in the L216S mutant. We also tested for genetic interactions between Ltv1_L216S and Rps15_RK and Rps15_YRR and observed epistasis with Rps15_YRR (Fig 7F), similar to the epistasis observed with the CTT-deletion of Ltv1 (Fig 7E). This finding further supports the conclusions above that (i) the Ltv1 CTT interacts with Rps15 via the YRR residues and (ii) that the CTT of Ltv1 is mispositioned in the L216S mutant. In contrast, both Tsr1_RK and Tsr1_Δloop as well as Rps15_RK are neutral to Ltv1_L216S (Fig 7G), indicating that the Ltv1_L216S mutation does not affect the interaction between Tsr1 and Rps15.

Thus, these genetic data demonstrate functional interactions between the CTTs of Ltv1, Rps15 and Tsr1, consistent with the cryo-EM structures. The DMS-MaPSeq data show that residues around the Tsr1 binding site are altered in ribosomes from cells with the LIPHAK mutation, indicating an alteration in this network during assembly. Intriguingly, many of the RNA residues with altered DMS accessibility in the L216S mutant are located at or near the hinge in Tsr1 that enables its movement from the decoding site helix to the beak as 80S-like ribosomes are formed [16]. Thus, the LIPHAK mutation seems to affect the movement of Tsr1 from the decoding site to the beak.

## Movement of Tsr1 to the beak is required for Ltv1 release

We have previously shown that Ltv1 phosphorylation is required for its dissociation [20, 21], but temporally separated, because it also required the phosphorylation of Rio2, as well as another, undefined step (Fig 1D, [21]). Importantly, our previous data showed that this step was blocked by mutations in Tsr1 (Tsr1_RK) and Rio2 (Rio2_loop, Rio2_K105E). Moreover, deletion of a long internal loop in Tsr1 (Tsr1_Δloop) rescues the Tsr1_RK mutation [21].

Many of the RNA residues of altered accessibility in the L216S mutant are located at or near a hinge in Tsr1, which allows it to swing away from the decoding site helix towards the beak, when pre-40S are joined by 60S subunits to form 80S-like ribosomes (Fig 7A and 7C, [16]). Importantly, given the location of the N-terminus of Ltv1 *under* Tsr1, this movement of Tsr1 is likely required not just for the formation of 80S-like ribosomes, but also the dissociation of Ltv1.

We therefore hypothesized that the unidentified step in the dissociation of Ltv1 was the hinge-driven movement of Tsr1 to the beak, and that this step was impaired in the Tsr1_RK, the Rio2_loop and Rio2_K105E mutants. To test this model, we wanted to determine if deleting the hinge in Tsr1 also blocked Ltv1 dissociation. Normally, this experiment would involve assessing the binding of Ltv1, Enp1 and Rio2 to pre-40S, and investigating Ltv1's phosphorylation status [21]. However, because Tsr1_Δhinge binds ribosomes weakly [16], all other AFs are also partially dissociated, as previously seen [16], we were unable to directly assess Ltv1 dissociation in sucrose gradients. Instead, we carried out an extensive genetic analysis, to demonstrate that deleting the hinge, which blocks Tsr1's movement [16], affects the same step affected by Tsr1_RK, Rio2_loop and Rio2_K105E, which is the dissociation of Ltv1 [21].

As described above, Tsr1_Δloop rescues Tsr1_RK [21], as well as Tsr1_Δhinge [16]. Thus, we tested if Tsr1_Δloop also rescued Rio2_loop and Rio2_K105E. Indeed, both mutations are rescued by this deletion in the Tsr1 internal loop, Tsr1_Δloop (Fig 7H). Importantly, Tsr1_Δloop also rescues mutation of the Ltv1 phosphorylation site, Ltv1_S/A (S6C Fig), demonstrating that it is linked to release of Ltv1. Moreover, Tsr1_RK, Tsr1_Δhinge and Rio2_K105E are all sick with Rps15_YRR (Fig 7I and [16], Rio2_loop was not tested). Thus, Tsr1_RK, Tsr1_Δhinge, Rio2_K105E and Rio2_loop have the same genetic interactions. We have previously shown that Tsr1_Δhinge blocks movement of Tsr1 to the beak [16]. Additionally, we have shown that Tsr1_RK, Rio2_loop and Rio2_K105E affect a yet unidentified step to activate the release of phosphorylated Ltv1 ([21], Fig 1B). Thus, taken together, these data suggest that the unidentified step that activates the release of Ltv1 is the movement of Tsr1 to the beak, as predicted by the structural data that indicate that Ltv1 is located under Tsr1.

## Blocking movement of Tsr1 to the beak impairs Rps31 recruitment

As described above, in ribosomes from cells expressing the Ltv1_L216S mutant most RPs are depleted from the 40S subunit head (Fig 5). Notably the protein that is most significantly depleted is Rps31. Rps31 is encoded as an N-terminal ubiquitin fusion protein, which is post-translationally cleaved into ubiquitin and Rps31. Previous data strongly suggest that it is incorporated into ribosomes as the fusion protein, as the unprocessed protein is found in polysomes when cleavage is delayed by a point mutation at the processing site [42]. Rps31 is a late-binding RP, which is not yet incorporated in the stable cytoplasmic 40S intermediates that have been visualized by cryo-EM [14, 15], although cryo-EM structures of later 80S-like ribosomes show it present ([16], Fig 1A). This is consistent with data demonstrating that depletion of Rps31 leads to accumulation of the cytoplasmic 20S rRNA [42]. Similarly, impairing the removal of the ubiquitin also delays Nob1-dependent cleavage in 80S-like ribosomes [42], as does deletion of the N-terminal extension of the protein [43]. Thus, the available structural

and biochemical data suggest strongly that Rps31 is incorporated around the transition between 40S and 80S-like ribosomes. Note that nonetheless yGFP-tagged Rps31 is found in the nuclei if 40S assembly is delayed [43]. Whether this reflects a weak association not captured by cryo-EM (*e.g.* by protein-protein interactions with Rps12), or is an artifact from the GFP fusion, or the stalled assembly is not clear.

Intriguingly, if Tsr1 were still located in its stable position near the decoding site, then the N-terminal extension of Rps31 would have to slide under Tsr1 (Fig 8A and 8B). Moreover, in that location, the head group of Tsr1 would clash with the N-terminal ubiquitin fusion (Fig 8A and 8B). Thus, the structures of pre-40S ribosomes suggest that Tsr1 blocks the recruitment of Rps31 to its binding site at the subunit interface, explaining why Rps31 recruitment is delayed until the formation of 80S-like ribosomes, which is accompanied by the hinge-motion-driven movement of Tsr1 ([16], Fig 8A and 8B). Together with the DMS-MaPseq data which demonstrate alterations around the Tsr1 hinge, these data thus suggest that the movement of Tsr1 to the beak is required for Rps31 recruitment.

To test this model, we investigated genetic interactions between Rps31 and Tsr1. If Tsr1_RK and Tsr1_Δhinge impair the movement of Tsr1 to the beak, and this movement is important for the incorporation of Rps31, then we expect that Tsr1_RK is rescued by deletion of the ubiquitin in Rps31_ubi. Indeed, Tsr1_RK shows small epistatic interactions with Rps31_ubi (Fig 8C).

In addition, previous work has indicated that removing an N-terminal tail in Rps31, which is eukaryote-specific and not conserved in archaea, which also do not conserve Tsr1, is required for assembly of Rps31 into ribosomes [43]. This tail stretches across the front of the head. Given that our data indicate a steric block for Rps31, we combined an Rps31 deletion mutant, lacking this stretch (but containing the ubiquitin), called Rps31_ΔN, with mutants in Tsr1. As seen before [43], Rps31_ΔN produces a strong growth defect. Whether this effect arises because now the ubiquitin is hindered even more by Tsr1, or because this stretch is required for an initial encounter complex, or both, remains unknown. Importantly, the growth defect from Rps31_ΔN is epistatic to Tsr1_Δloop (Fig 8D), with a small but not statistically significant growth rescue. In contrast, Rps31_ΔN is synthetically sick with Tsr1_RK (Fig 8E).

To further support the model that Ltv1 affects Rps31 incorporation via Tsr1, we combined the Ltv1_L216S mutation with deletion of the ubiquitin in Rps31, Rps31_Δubi. If Ltv1_L216S blocks assembly upstream of the movement of Tsr1 to the beak, then removing the ubiquitin-portion of the gene, which cannot be accommodated without Tsr1 movement, should partially rescue the defects from the L216S mutation. Indeed, while the ubiquitin deletion has no effect on growth in the background of wt Ltv1, it slightly but reproducibly rescues the growth defect of the Ltv1_L216S mutation (Fig 8F).

Together, these data provide strong evidence that indeed movement of Tsr1 towards the beak is required for the incorporation of Rps31, and that this movement is impaired by the Ltv1_L216S mutation, thereby explaining the reduced incorporation of Rps31.

## Discussion

### Ltv1 orchestrates the hierarchy of head RP binding

Taking advantage of a novel mutation in Ltv1, Ltv1_L216S, which is associated with a human dermatological condition known as LIPHAK disease [35], we have uncovered here sweeping roles for Ltv1 in orchestrating the assembly of the small ribosomal subunit head, using a number of different strategies as discussed below.

Previous work has shown that Ltv1 directly binds Rps3/uS3, Rps15/uS19 and Rps20/uS10, and is required for the efficient recruitment of Asc1 and Rps10 [24,26,32–34]. Moreover,

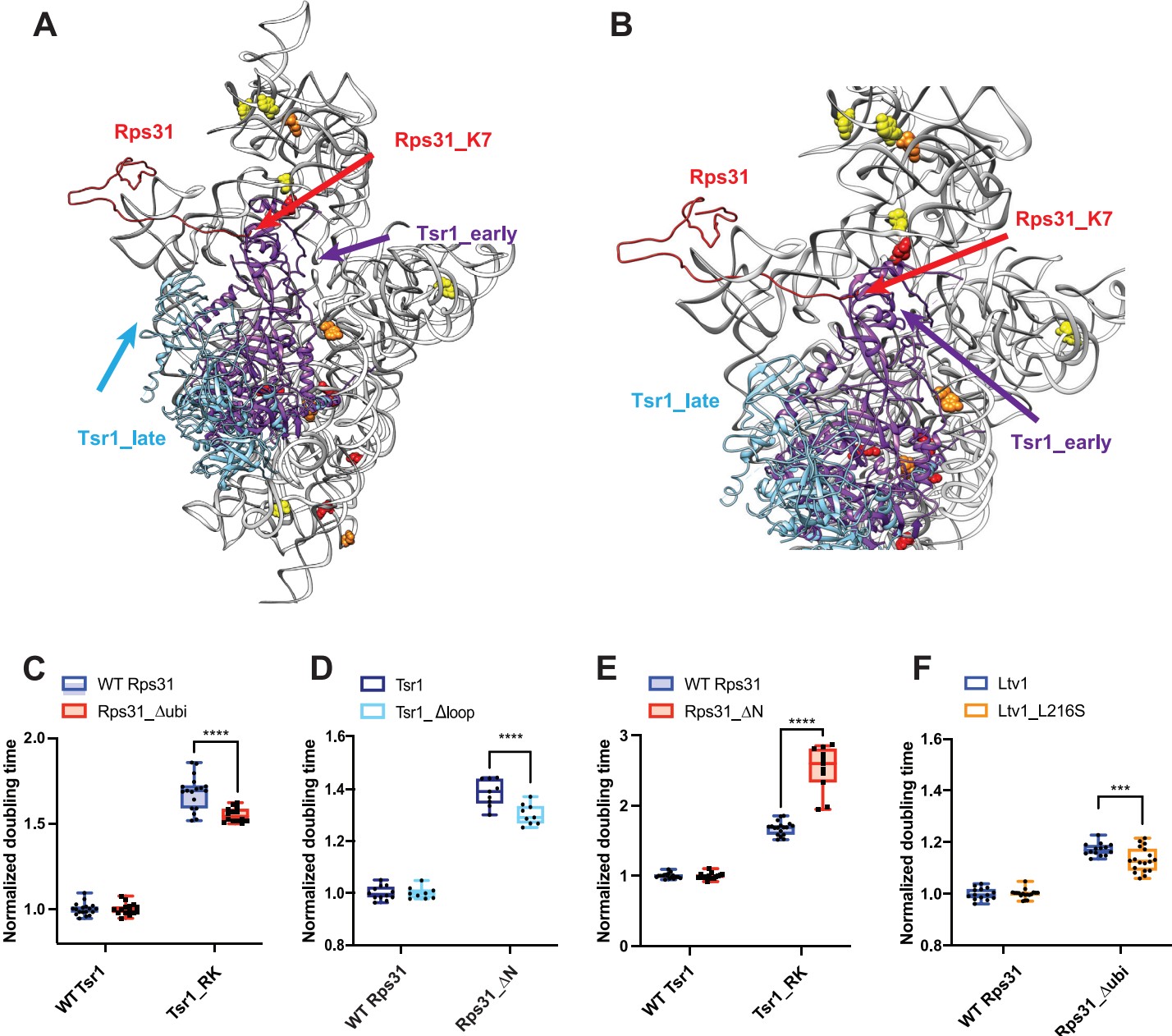

**Fig 8. Rps31 binding requires Tsr1 movement to the beak.** (A) Composite structure of pre-40S ribosomes (6FAI) with Tsr1 in the early (6FAI) and late position (6WDR) and showing Rps31 from mature ribosomes (3JAM). 6FAI and 3JAM ribosomes were overlaid by matching Rps18, and 6WDR was overlaid onto these by matching 18S rRNA. (B) Structural detail of a top view of the structure in panel A highlighting the N-terminus of Rps31 near Tsr1 in the early position. Note that residues 1–6 are not resolved, the first residue visible is lysine 7 (K7). (C) Doubling times (normalized to wt Tsr1) for yeast cells expressing either wt Rps31 or Rps31 lacking the N-terminal ubiquitin fusion (Rps31Δubi), and either wt Tsr1, or Tsr1_RK. (D) Doubling times (normalized to wt Rps31) for yeast cells expressing either wt Rps31 or Rps31 lacking the N-terminal extension (Rps31ΔN), and either wt Tsr1, or Tsr1_Δloop. (E) Doubling times (normalized to wt Tsr1) for yeast cells expressing either wt Rps31 or Rps31 lacking the N-terminal extension (Rps31ΔN) and either wt Tsr1, or Tsr1_RK. (F) Doubling times (normalized to wt Rps31) for yeast cells expressing either wt Rps31 or Rps31Δubi, and either wt Ltv1, or Ltv1_L216S. Significance for panels C-F was tested using a two-way ANOVA. ***, $P < 0.001$; ****, $P < 0.0001$.

previous data suggested that Rps3 positioning is affected directly by Ltv1, thereby mediating positioning of Rps17 and Asc1 [26]. Finally, roles in quality control of head assembly have also been described, in particular proofreading of the positioning for Rps3/uS3, Rps15/uS19, and Rps20/uS10 [21]. Here, we expand these roles to show that Ltv1 also directly binds Rps12 and Rps29/uS14 and orchestrates the recruitment of Rps12, Rps29/uS14 and Rps31. For Rps12 and Rps29/uS14 this is consistent with previous genetic data [26,39]. Moreover, our data also indicate that correct positioning of Rps15/uS19 depends on Ltv1. Thus, Ltv1 directly binds 5 of the 15 head binding RPs (Fig 9A) and is involved in the assembly of 9 of them (Fig 9B), thereby explaining the global defects in head assembly that we observe here in ribosomes from cells containing the LIPHAK mutation. Notably, 4 of the 5 RPs bound by Ltv1 are universally conserved, which is surprising, given it is a eukaryote-specific protein. However, the additional 5 proteins who depend on Ltv1 for their incorporation are all eukaryote-specific. This suggests that Ltv1 enables the assembly of the eukaryote-specific ribosome by modulating the binding of universally conserved RPs, as described in further detail below.

More importantly, the data herein demonstrate for the first time how Ltv1 carries out its roles. By directly binding Rps12 and Rps29, Ltv1 may help in their recruitment. Moreover, the direct interaction with Rps3, Rps15 and Rps20 enables proper positioning of these RPs, as shown previously for Rps3 and here for Rps15 and Rps20. More surprisingly, the data also demonstrate that by binding to Rps3 and Rps20 directly, Ltv1 prevents the premature formation of their contact, thereby explaining previous genetic and biochemical data [26]. The data in here show that the LIPHAK mutation mispositions Ltv1, thereby dislodging the protein from Rps20 and allowing for the premature formation of the Rps3/Rps20 contact. Importantly, this appears to also misposition Rps3 and Rps20, as the LIPHAK cells are hyper-sensitive to mutations in the Rps3/Rps20 contact. Moreover, we have previously shown that in cells lacking Ltv1 entirely, their interaction partners Rps10 and Asc1 are substoichiometrically incorporated into ribosomes [26]. Thus, Ltv1 uses its direct binding interactions with 5 universally conserved head RPs to order their assembly, both by prioritizing some and delaying others.

The observation that Ltv1 blocks the formation of the contact between Rps3 and Rps20 also suggests that Ltv1 must move to enable the formation of this contact and allow for its phosphorylation by Hrr25 [21], indicating that flexibility in Ltv1 might be functionally important.

In addition to using direct binding interactions to orchestrate the RPs assembly hierarchy of the head, Ltv1 also affects recruitment of two RPs indirectly via conformational transitions in the nascent 40S subunit. Our analysis of previous structures of assembling ribosomes indicates that Rps20 blocks access of Rps29 both directly, as it is located on top of Rps29, as well as directly, by promoting folding of h39 and h41, which further restricts access of Rps29 to its binding site (S5 Fig). Thus, Rps29 must bind before Rps20 enables the docking of h39 (Fig 9C), as we have confirmed with genetic data, and consistent with previous biochemical data [26]. The premature formation of the Rps3/Rps20 contact will therefore prematurely enable the binding of Rps20 and the stabilization of h39 in its tertiary contact and is therefore expected to impair the recruitment of Rps29, which we confirm using ribosome purification and genetic analyses. Thus, Ltv1 enables the recruitment of Rps29 by delaying the binding of Rps20 and the formation of rRNA tertiary structure.

In addition, our data also demonstrate how Ltv1 orchestrates the recruitment of Rps31. Our genetic data strongly suggest that Rps31 binding is blocked by Tsr1, which has to rotate towards the beak to enable Rps31 binding and Ltv1 dissociation (Fig 9D). This finding is consistent with existing cryo-EM structures that show Rps31 binding occurs with the Tsr1 movement to the beak. Our data show that Ltv1 enables this rotation via contacts between its C-terminal tail, the C-terminal tail of Rps15 and Tsr1 itself. Moreover, our data also strongly

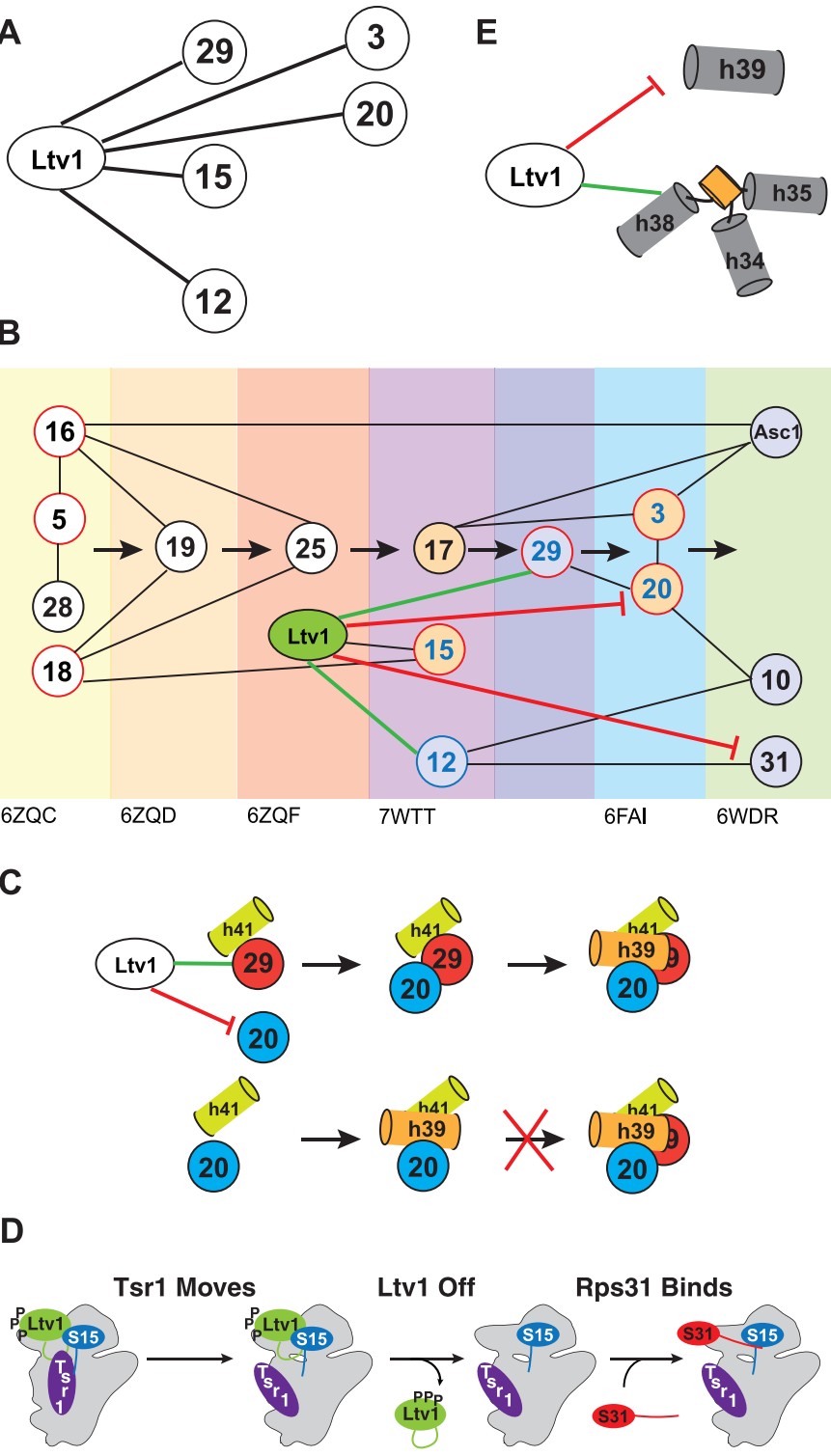

**Fig 9. Role of Ltv1 in head assembly.** (A) Direct binding interactions for Ltv1. (B) Ltv1 orchestrates the hierarchy of head assembly. Ltv1 is recruited around the same time as Rps25 binds, as determined by the appearance of the C-terminal extension in human ribosomes. RPs that bind directly to Ltv1 are shown in blue circles. RPs Ltv1 helps recruit are connected by a green line and proteins delayed by Ltv1 are connected by a red line. Proteins that are reduced or mispositioned when Ltv1 is absent or mutated are shown in blue or orange spacefill, respectively. Universally conserved RPs are shown in a red circle. (C) Ltv1 orchestrates the order of binding of Rps20 and Rps29. By recruiting Rps29 and blocking. The binding of Rps20, Ltv1 enables Rps29 to bind before Rps20, whose binding stabilizes helix 39

(h39) in its tertiary contact with helix 41 (h41). Without Ltv1 (or with the Ltv1_L216S mutation) Rps20 binds first, thus stabilizing h39 and blocking access for Rps29. (D) Tsr1 movement to the beak is required for Ltv1 dissociation and Rps31 binding. Parts of Ltv1 (illustrated by the green loop) bind under Tsr1, requiring its movement for dissociation. This also allows for recruitment of Rps31. Tsr1 movement is regulated via Rps15 and Ltv1. (E) Ltv1 delays formation of h39 and facilitates the formation of j34-35-38.

suggest that this movement of Tsr1 towards the beak is required for Ltv1 release, recognizing it as a previously identified activation step for Ltv1 release [21].

Ltv1 dissociation is required for Enp1 binding [21], which is explained by the observation that Enp1 is buried under Ltv1 [14,15]. Moreover, cryo-EM structures demonstrate that Enp1 blocks the binding of Rps10 at its mature site [25]. Thus, Rps10 binding requires Enp1 dissociation, which itself demands Ltv1 release. Thus, Ltv1 also indirectly regulates the timing of Rps10 incorporation.

These observations are summarized in Fig 9B, which illustrates the central role of Ltv1 in head assembly and explains much of the previously unexplained parts of the assembly hierarchy. Close inspection of available cryo-EM structures indicates that Ltv1 is recruited to rRNA posttranscriptionally around the time of Rps25 recruitment and after binding of Rps19, as its C-terminal helix at the subunit interface becomes visible then [44]. Ltv1's binding interactions with Rps15 and Rps12 explain how these are then recruited in the next assembly step, even though their RP interaction partners have been unchanged. The next available structure shows recruitment of Rps3, Rps20 and Rps29, but as described above, the data in here show that Rps29 is recruited first, both via directly binding to Ltv1, and because Ltv1 delays Rps20 recruitment, consistent with structures that show Rps29 being buried once Rps20 is bound and its interacting helix, h39, is docked. Future experiments will be required to dissect how Ltv1 and/or Rps20 are moving to enable the formation of the contact between Rps20 and Rps3, which is required for Ltv1 phosphorylation and release [21]. Finally, the data also explain why Rps31 binding is delayed, even though its only interaction partner, Rps12, is bound earlier.

## Ltv1 facilitates rRNA folding

In addition, our data also indicate that Ltv1 plays direct roles in chaperoning the folding of j34-35-38 (Fig 9E). We have previously shown that this 3-helix junction folds upon Ltv1 dissociation, indicating a role for Ltv1 in delaying its folding [14,15,21,22]. The data in here strongly suggest that Ltv1 binds this junction directly and demonstrate that this interaction is required for proper folding of j34-35-38. Interestingly, mapping residues that are altered *in vivo* between a Tsr1-TAP and a Nob1 pulldown [45], which essentially corresponds to intermediates prior to and after dissociation of Ltv1, demonstrates changes around j34-35-38, directly adjacent to where our structure predicts Ltv1 to bind, as well as at several other spots predicted to be directly contacted by Ltv1 (S7 Fig), providing additional validation for the placement of Ltv1 and providing a nice illustration of how folding of j34-35-38 affects the subunit head. While future experiments will be required to better understand how this (and other) 3-helix junctions fold, and how Ltv1 blocks the formation of its structure, the observation that disrupting h34 and h35, which lead into the junction, exacerbates effects from the LIPHAK mutation indicates that Ltv1 does not disrupt these helices. Instead Ltv1 may bind the connecting elements. Indeed, overlaying an earlier 40S structure indicates that h34 is formed prior to h35 and h38, but not yet positioned correctly. Interestingly, one of the strands of h35 binds directly to Ltv1, suggesting that Ltv1 might function in folding of h34-35-38 either by positioning h35 or delaying the formation of this short helix that emanates from the junction. Future experiments will address how folding of 3 helix junctions can be facilitated.

We have previously shown that Rio2 plays similar roles in delaying the formation of a tertiary contact between the loop of h31 and j34-35-38 [22]. Together, these findings indicate the importance of a carefully orchestrated rRNA folding order to enable efficient ribosome assembly. While Rio2 appears to directly block the formation of a tertiary contact, Ltv1 appears to do this indirectly, by blocking the assembly of an RP, which stabilizes this tertiary contact.

In addition to chaperoning the folding of j34-35-38, our data also strongly suggest roles for Ltv1 in folding of h39, and its tertiary contact with h41 (Fig 9E). The structure in here, together with previous biochemical data indicate that Ltv1 blocks the formation of the Rps3-Rps20 contact, thereby delaying the recruitment of Rps20, which binds and stabilizes the docking of h39 into its tertiary interaction. Our previous observation that residues at the tip of h39 are more protected in ribosomes from cells lacking Ltv1 [26], indicating their misfolding, suggest that by delaying this contact between h39 and h41, Ltv1 also chaperones the folding of this helix.

### The cancer-associated Rps15-S136F mutation enables unregulated Tsr1 rotation

We have previously shown that the Tsr1_RK mutation can be bypassed by a mutation in Rps15 that is associated with chronic lymphocytic leukemia (CLL), Rps15_S136F, which leads to defects in the resulting ribosome population [21]. However, it was unclear what step exactly was blocked by the Tsr1_RK mutation. The data in here strongly suggest that the step blocked by the Tsr1_RK mutation is the rotation of Tsr1 towards the beak. Interestingly, Rps15_S136F does not rescue Tsr1_Δhinge, indicating that the Rps15 mutation does not affect the movement *per se* but its regulation via the Ltv1/Rps15/Tsr1 network. Given that Tsr1_Δloop rescues all the mutants that block the hinge movement, we speculate that the loop that is removed by the Tsr1_Δloop mutation holds Tsr1 near the beak, via interactions with Rps15, as shown in S5A Fig. Thus, the cancer-associated Rps15_S136F mutation, which leads to increased misinitiation at non-AUG codons, may perturb Rps15 directly, or affect how Ltv1 (or other, yet undescribed factors) regulates the interaction between Rps15 and the loop in Tsr1.

## Materials and methods

### Yeast strains and plasmids

*Saccharomyces cerevisiae* strains (S2 Table) were either obtained from the Horizon Dharmacon Yeast Knockout Collection, or created by homologous recombination [46] and confirmed by serial dilution, PCR, and Western blotting in cases where antibodies were available. Plasmids were generated by standard cloning techniques and confirmed via Sanger sequencing, and are listed in S3 Table.

### *In vivo* subunit joining assay

Gal:Fap7 cells (with the appropriate additional genomic alterations) were grown at 30˚C for >16h in YPD to deplete Fap7, and harvested in the presence of cycloheximide as described [21,36,47]. Cells were lysed under liquid $N_2$, cell debris spun out and the supernatant loaded onto a sucrose gradient as described below.

### Sucrose-gradient fractionation

5,000–7,500 OD of clarified lysate (~ 200 μl) were loaded onto a 10–50% sucrose-gradient and centrifuged at 40,000 rpm for 2 hours using a SW41Ti rotor before fractionation into 700 μl fractions. Northern blot samples were prepared by phenol chloroform extraction on 200 μl of each fraction. For Western blotting, fractions were mixed with SDS loading buffer, and ran on

an SDS-PAGE gel, transferred and probed with the appropriate antibodies, as explained in [21, 36, 47].

## Northern blot

Northern blots were prepared from ΔLtv1 strain supplemented with either Ltv1, or Ltv1_L216S. The cells were grown overnight in minimal media, diluted into fresh YPD and harvested. Total RNA was isolated through phenol/chloroform RNA extraction, denatured in formaldehyde, and separated on an agarose gel. The following probes were used: 20S: GCTCTCATGCTCTTGCC; 18S: CATGGCTTAATCTTTGAGAC; 25S: GCCCGTTCCCTTGGCTGTG; and U2: CAGATACTACACTTG.

## Growth curve measurements

Cells were grown in glucose minimal media or YPD overnight, diluted into fresh minimal media or YPD for 3–6 hours, before diluting them into a 96-well plate at a starting OD of 0.04–0.1 and placed in Synergy.2 plate reader (BioTek) to record $OD_{600}$ for 48 hours while the plate was shaking at 30˚C.

rRNA mutants were grown in minimal media or YPD at 30˚C overnight and diluted into fresh YPD to incubate for 3h at 30˚C before changing the temperature to 37˚C for 3h. Cells were then inoculated into YPD in a 96-well plate, where the growth was recorded for 48 hours at 37˚C.

## Ribosome purification

Cells were resuspended in ribosome lysis buffer (20 mM Hepes/KOH at pH 7.4, 100 mM KOAc, and 2.5 mM $Mg(OAc)_2$) supplemented with 1 mg/ml heparin, 1 mM benzamidine, 2 mM DTT, and protease inhibitor cocktail (Sigma-Aldrich) and lysed under liquid $N_2$. The cell lysates were thawed and clarified before layering 200 μl onto 500 μl sucrose cushion and centrifuging at 70k rpm for 65 min in a Ti110 rotor. The supernatant was discarded, and the pellet containing the ribosomes was resuspended in high salt buffer (ribosome buffer, 500 mM KCl, 1 mg/ml heparin, and 2 mM DTT) and layered onto another 500 μl sucrose cushion. The tubes were centrifuged at 100k rpm for 75 min, and the pelleted ribosomes were resuspended in subunit separation buffer (50 mM Hepes/KOH at pH 7.4, 500 mM KCl, 2 mM $MgCl_2$, 2 mM DTT and 1 mM puromycin (Sigma-Aldrich), incubated for 1h at 37˚C, and loaded onto a 5–20% sucrose gradient (50 mM Hepes/KOH, pH 7.4, 500 mM KCl, 5 mM $MgCl_2$, 2 mM DTT and 0.1 mM EDTA) and centrifuged for 8 h at 30k rpm. To isolate the separated subunits the gradients were fractionated and 40S subunits collected, buffer exchanged into ribosome storage buffer (ribosome buffer with 250 mM sucrose and 2 mM DTT) and concentrated to be flash frozen and stored at -80˚C for subsequent mass spectrometry or Western analysis.

## Mass spectrometry

Purified ribosomes were prepared as stated above and precipitated in acetone. Three biological replicates of WT Ltv1 and Ltv1_L216S were analyzed. Raw data were deposited to the PRIDE database under accession number PXD046239.

## DMS MaPseq sample preparations

Prior to flash freezing the purified ribosomes were treated with or without 1% DMS (Sigma-Aldrich) in the presence of 80 mM Hepes pH 7.4, 50 mM NaCl, 5 mM $Mg(OAc)_2$, and 0.2 μM RNaseP RNA at 30˚C for 5 min. DMS reactions were stopped by the addition of 0.4 volumes

stop buffer (1M β-ME, 1.5M NaOAc, pH 5.2), and purified using phenol chloroform precipitation.

## DMS MaPseq RNA library preparation

RNA library preparation was adapted from [48] and performed as described [22, 41]. Briefly, RNAs were fragmented with the addition of 20 mM $MgCl_2$ at 94°C for 10 min. Fragments were separated on a denaturing 15% gel, and fragments between 50–80 nt were cut out and RNA was eluted overnight in RNA gel extraction buffer (0.3 M NaOAc, pH = 5.5, 1 mM EDTA (ethylenediaminetetraacetic acid), 0.25% vol/vol SDS (sodium dodecyl sulfate). Eluted RNA fragments were dephosphorylated using T4 PNK (NEB), ligated to an adaptor using T4 RNA ligase II truncated KQ(NEB), and then gel purified. TGIRT III (InGex) was used to reverse transcribe RNAs in 50 mM Tris HCl, pH = 8.3, 75 mM KCl, 3 mM $MgCl_2$, 1 mM dNTPs, 5 mM DTT, and 10 U SUPERase·In (Invitrogen) for 1.5 h at 60°C. The RNA was later hydrolyzed by 250 mM NaOH and the cDNA was circularized using CircLigase II (Lucigen). The circularized cDNA was shipped to the Scripps La Jolla Genomics Core for final library preparation with Illumina sequencing adaptors and sequenced using single-end sequencing on the Illumina NextSeq 500 platform.

## DMS MaPseq RNA data processing

DMS-MaPseq data processing followed a GitHub pipeline (https://github.com/borisz264/mod_seq). The adaptor sequence (GATCGGAAGAGCACACGTCTGAACTCCAGTCA) was trimmed from raw sequence via Skewer [49]. The first 5 nt, last 5 nt, and low quality nt were trimmed by Shapemapper 2.0. Reads were aligned to Saccharomyces cerevisiae 20S rRNA (Saccharomyces Genome Database), and read coverage and mutations at each position were counted by Shapemapper 2.0 to obtain the mutational rate at each nucleotide.

The DMS accessibility for each individual A or C nucleotide was calculated by dividing the mutational rate for each A and C by the average mutational rate for all untreated A or C nucleotides, respectively. Thereafter, the values from the untreated samples were subtracted from the values for the DMS-treated samples, to obtained normalized DMS accessibility values for each nucleotide. Finally, the normalized DMS accessibility values from the two biological replicates were averaged.

Each residue was then sorted into one of four bins according to its normalized DMS accessibility score: 0.0–0.5, 0.5–1.0, 1.0–2.0, 2.0–4.0, and above 4. Residues that changed by one or more bins were considered to have different DMS accessibility.

Residues that are changed by one bin are indicated in yellow, residues changing one bin in one sample and two bins in one sample are shown in orange, and residues changing two bins in two samples are indicated in red.

## Antibodies

Antibodies against AFs were raised by Josman LLC in rabbits against purified recombinant protein, tested against recombinant protein and yeast cell lysate. Antibodies against Rps10 and Rps26 were raised in rabbits against chemically synthesized peptides by New England Peptide and tested against yeast cell lysate and recombinant protein (Rps26 only). The HA-antibody was from BioLegend.

## Supporting information

**S1 Fig. Yeast Ltv1_L216S corresponds to human Ltv1_N168S. (relates to Fig 2).** (A) Sequence alignment of a portion of Ltv1 highlighting N168 and L216 in human and yeast Ltv1, respectively. (B) Doubling times of wt yeast cells (BY4741), or ΔLtv1 cells either expressing Ltv1 from a TEF-promoter-driven plasmid (WT Ltv1), or containing and empty vectors (EV). (C) Normalized doubling time of yeast lacking endogenous Ltv1 and expressing either wt Ltv1, Ltv1_L216S or Ltv1_L217S from TEF-promoter-driven plasmids. Note that the data for WT Ltv1 and Ltv1_L216S are the same as in Fig 1B. (D) Normalized doubling time of yeast lacking endogenous Ltv1 and expressing either wt Ltv1, Ltv1_L216S or Ltv1_L217S from either TEF or Cyc1-promoter-driven plasmids. Significance in panels B-D was tested using a two-way ANOVA. ***, $P<0.001$; ****, $P<0.0001$. (E) Left: Western analysis of yeast lysates prepared from cells lacking endogenous Ltv1 and expressing either wt Ltv1 or Ltv1_L216S from TEF-promoter-driven plasmids. Three replicates for each are shown, as well as a control from ΔLtv1 cells. Fap7 is used as a loading control. Right: Quantification of the data on the left.
(EPS)

**S2 Fig. Enp1_WKK binds Ltv1 weakly, allowing for its phosphorylation independent release. (relates to Fig 2).** (A) Doubling time of yeast depleted for endogenous Enp1 and lacking endogenous Ltv1 and expressing either wt Enp1 or Enp1_WKK and wt Ltv1 or phosphorylation-deficient Ltv1_S/A from TEF-promoter-driven plasmids. Significance was tested using a two-way ANOVA. ****, $P<0.0001$. (B) Absorbance profile (top) and Western blot (bottom) of yeast lysates from cells expressing either wt Enp1 or Enp1_WKK. (C) Doubling time of yeast depleted for endogenous Enp1 and lacking endogenous Ltv1 and expressing either wt Enp1 or Enp1_WKK and wt Ltv1 or Ltv1_L216S from TEF-promoter-driven plasmids. Significance was tested using a two-way ANOVA. ****, $P<0.0001$.
(EPS)

**S3 Fig. Enp1_WKK binds Ltv1 weakly, allowing for its phosphorylation independent release. (relates to Fig 4).** Doubling times for yeast cells containing a temperature sensitive RNAPol I mutation (NOY504) and expressing plasmid encoded wt or mutant 18S rRNA and wt Ltv1. Significance was tested using a two-way ANOVA. **, $P<0.01$; ****, $P<0.0001$.
(EPS)

**S4 Fig. DMS-MaPseq quality control. (relates to Fig 6).** (A) Mutational rate changes for A and C residues upon DMS addition. Note that G also does get modified by DMS. (B) High read depth over the entire molecule. (C) Mature 18S rRNA is captured in all samples.
(EPS)

**S5 Fig. Binding of Rps20 is coupled to folding of h39 (relates to Fig 6).** (A) Structural detail of the 40S head (from PDB ID 7WTT) before Rps20 binding. For clarity only Rps29 is shown. (B) Structural detail of the 40S head before (blue, PDB ID 7WTT) and after Rps20 binding (silver, PDB ID 6FAI). For clarity only Rps29 is shown. Access for Rps29 is compromised after the folding of h39 and the h41 bulge, induced by Rps20 binding. (C-D) Rps29 (red) is located under Rps20 (cyan). Structures from PDB ID 7WTT and 6FAI, respectively. (E) Schematic for effects on Rps20, or Rps29 recruitment expected from the EYER mutation, when Rps29 binds before Rps20 (top) or vice versa (bottom).
(EPS)

**S6 Fig. A structural and genetic network between Ltv1, Rps15 and Tsr1 (relates to Fig 7).** (A) Structural detail of a composite structure of yeast pre-40S (PDB:6FAI) and human Ltv1

(PDB ID 6G18) highlighting elements in Tsr1, Rps15 and Ltv1. The residues mutated in Rps15_YRR and Rps15_RK are shown in blue and cyan spheres, respectively. The residues mutated in Tsr1_RK are shown in magenta spheres, and the first and last amino acids of the loop removed in Tsr1_Δloop are shown in magenta. The last amino acid in the Ltv1 truncation Ltv1_1–394 is shown in green space fill. (B) Doubling times for yeast cells expressing either wt Ltv1 or Ltv1_3–194, and Tsr1_RK_Δloop. Data for WT Ltv1 and Tsr1_RK_Δloop are in [21]. (C) Doubling times for yeast cells expressing either wt Ltv1 or the phosphorylation-deficient Ltv1_S/A and either wt Tsr1 or Tsr1_Δloop. (D) Doubling times (normalized to wt Rps15) for yeast cells expressing either wt Rps15, Rps15_YRR or Rps15_RK and wt Tsr1 or Tsr1_RK. (E) Doubling times (normalized to wt Rps15) for yeast cells expressing either wt Rps15, Rps15_YRR or Rps15_RK and wt Ltv1 or Ltv1_1–394. Significance for panels C-E was tested using a two-way ANOVA. ***, P<0.001; ****, P<0.0001. (EPS)

**S7 Fig. Ltv1 dissocation affects rRNA structure near Ltv1 binding sites.** Residues with altered SHAPE reactivity between intermediates enriched with Tsr1 and Nob1 [45] are mapped onto the predicted Ltv1-bound pre-40S. Ltv1 is shown in magenta, Rps20, with the residues D113/E115 (Rps20_DE) in cyan. Residues exposed in the Nob1 but not the Tsr1-bound structure (after Ltv1 dissociation), are shown in red, and residues protected in that transition are shown in green space-fill. (EPS)

**S1 Table. Residues with altered DMS accessibility.** (DOCX)

**S2 Table. Yeast strains used in this study.** (DOCX)

**S3 Table. Plasmids used in this study.** (DOCX)

**S1 Data. Supplementary data.** (XLSX)

## Acknowledgments

We thank John McGrath (King's College London) for providing us with information of the L216S mutation, and members of the Karbstein lab for discussion and comments on the manuscript.

## Author Contributions

**Conceptualization:** Katrin Karbstein.

**Formal analysis:** Ebba K. Blomqvist.

**Funding acquisition:** Katrin Karbstein.

**Investigation:** Ebba K. Blomqvist, Haina Huang, Katrin Karbstein.

**Project administration:** Katrin Karbstein.

**Resources:** Haina Huang.

**Supervision:** Katrin Karbstein.

**Writing – original draft:** Ebba K. Blomqvist, Katrin Karbstein.

**Writing – review & editing:** Ebba K. Blomqvist, Haina Huang, Katrin Karbstein.

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
