## [Decision Letter · Decision Letter 0]

19 Aug 2023

Dear Dr. Karbstein,

Thank you very much for submitting your Research Article entitled 'A disease associated mutant reveals how Ltv1 orchestrates RP assembly and rRNA folding of the small ribosomal subunit head' to PLOS Genetics.

Your manuscript was fully evaluated at the editorial level and by three independent peer reviewers. I am delighted to inform you that the three reviewers agree that you have submitted a well-conducted study that advances the knowledge of the 40S ribosome subunit assembly and possibly leads to an understanding of a human disease (but see comments of Reviewer 3 regarding whether he human Ltv1 N168S is analogous to yeast Ltv1 L216S). None of the reviewers requests additional data, but each provides thoughtful comments regarding alternative interpretation of some of the data and as well as suggestions for improving the clarity and presentation of your manuscript.Therefore, we ask you address these concerns/comments in a revised manuscript. Your revisions should address the specific points made by each reviewer.

We thank you for submitting this nice study to PLOS Genetics. We hope to receive your revised manuscript within the next 30 days. If you anticipate any delay in its return, we would ask you to let us know the expected resubmission date by email to plosgenetics@plos.org.

Yours sincerely,

Anita K. Hopper

Academic Editor

PLOS Genetics

Gregory P. Copenhaver

Editor-in-Chief

PLOS Genetics

Reviewer's Responses to Questions

**Comments to the Authors:**

Reviewer #1: This manuscript by Blomqvist et al. focuses on the role(s) of the ribosome assembly factor Lvt1 in 40S subunit maturation. Using yeast as a model, the authors primarily use ΔLtn1 cells and express WT or disease mutant (L216S) to determine affects on 40S maturation. The major hypothesis is that the Lvt1 L216S mutant is mispositioned on the 40S ribosome which is tested by a multitude of genetic mutants that support this hypothesis. The authors overlay the pre-40S cryo-EM and Lvt1 Alphafold structures. The position was confirmed by mutating specific putative contacts and observing synergistic or epistatic genetic interactions with yeast doubling time as a readout. The L216S mutant was positioned on the Head substructure of the pre-40S, which is highly supported by the loss (mass spec of purified 40S subunits) of primarily head subdomain RPs, but not RPs of the Platform or Body. DMS MaPSeq also demonstrated 18S rRNA changes in the Head and elsewhere, with the ladder aiding to explain downstream maturing defects. In total, the manuscript is strong, novel, and the conclusions are supported by genetic and/or biochemical evidence.

My only stronger/major comments are:

1) It would be worth pointing out to readers whether or not expressing Lvt1 on plasmids in stains that lack endogenous Ltn1 fully or partially rescues doubling time. I don’t feel this is a big weakness as the authors nicely show the difference between WT and L216S. But it would be informative to the readers. The polysomes for the Lvt1 strain in Figure 2B (left) still has a small 40S peak suggesting there is still a 40S maturation defect.

2) At times the nomenclature for the mutants goes back and forth of being very clear of the specific mutation (e.g., Ltv1_L216S; Rio2_K105E; S31_ ΔN) to being a bit confusing (e.g., Tsr1_RR for the R709E, K712E mutations; Ltv1_S/D for the S336D, S339D, S342D mutations. Table S3 does help with this but it is not intuitive as you read through the text. However, I can appreciate many of the mutants have already been published so changing those names would not be recommended.

3) For Figure 4F, the authors write “…demonstrate strong synthetic sick effects”. The legends notes for panel F, “Significance was tested using an unpaired t-test. ****, P<0.0001” but figure does not have markings (****) above the data to highlight this statistical test. Given that the standard deviation in these data sets must be pretty large, I am not convinced that a t test would demonstrate such a low p value or a difference that is statistically significant. Do these data accurately represent a strong effect?

Other minor comments/suggestions/typos:

4) At times, the data are presented differently, which makes it hard to follow at times. For example, Figure 2D, some data are presented in box plots and some data are presented as (what I think are) the median and range. I think this is due to some groups having 3 data points and some having >3 data points. If other reviewers also comment on this, it may more informative and more straightforward to represented the data with the individual points but include the mean as the bar/line and error bars showing SD or SEM.

5) On page 7, second paragraph, the authors state they tested whether Ltv1 chaperons the folding of j34-35-38 or indirectly affects its folding by preventing the premature folding of 131, or both. However, the subsequent text states the indirect effect would be inferred by an epistatic result and the direct effect would be inferred by a synergistic result. What would one expect if it is both (as the authors suggested was tested)?

6) On page 7, “The structure shows that Ltv1 also contacts 131 (Figure 4B), and mutation of residues Y82, D83, and Y84, which are adjacent to 131 to VAV, to make Ltv1_YDY produces a small growth defect (Figure 4C), therefore further validating the structure.” This sentence is very hard to follow in the current form.

7) Figure 5C is first discussed after Figure 6A-G.

8) Is the structure of Tsr1 also from Alphafold? Is its positioning on the pre-40S also made through molecular docking (as was done for Ltv1) or is it empirically determined?

9) In Figure 7E, the mutant is written as “Ltv1_1-394” on the x axis but written as “Ltv1_C394” in the text.

10) In Figure 7F and G, the text is on the x axis is squished.

11) In Figure 7H, the mutant is written as “Rio2_Nloop” on the x axis but written as “Rio2_loop” in the text and legend.

12) In Figure 8C-F, different sized asterisks/stars are used to denote statistical significance. I believe the legends is also missing “***, P<0.001.” for panel F.

Reviewer #2: In this work, the authors examine the role of the ribosome assembly factor, Ltv1 using a yeast model system. Ltv1 is known to be involved in assembly of the pre-40S head domain. Previous work demonstrated that Ltv1 helps recruit r-proteins, is implicated in helping rRNA folding, and is involved in quality control during 40S maturation. Here, the authors use a disease-relevant mutation, Ltv1_L216S, which exhibits a modest growth phenotype, to characterize the functions of Ltv1. The authors begin by demonstrating that Ltv1_L216S mutation impairs 80S-like ribosome formation and inhibits Ltv1 phosphorylation, which suggests that Ltv1_L216S is mispositioned on 40S and is a useful marker for dissecting the genetic interaction networks involving Ltv1.

The authors use structural modeling and targeted mutational analyses to model Ltv1 structure in the pre-40S head domain. Strikingly, the structure predicts that Ltv1 is extended along the majority of the head domain, interacting directly with Rps3, Rps15, Rps20, Rps29, and Rps12 and h34. They carry out mass spectrometry analysis on the Ltv1_L216S ribosomes and revealed that recruitment of many RPs is impaired indicating that Ltv1 positioning is critical for RP recruitment. The authors also performed DMS-MapSeq and uncovered multiple clustered sites of altered reactivity supporting global mispositioning of Ltv1_L216S on the ribosome. Guided by the structure probing, the authors use genetic analyses to uncover that premature contact between Rps20 and Rps3 may prevent binding of other RPs, specifically Rps29. They also showed that altered DMS reactivity near the subunit interface is likely due to disruption of an interaction network between Tsr1 and Rps15. Finally, the authors show that mutations in the hinge region of Tsr1 causes the same genetic interactions with other mutations that influence Ltv1 release and that Tsr1 hinge mutations also lead to impaired recruitment of Rps31. Taken together their data suggests a mechanism in which Tsr1 movement is required to release Ltv1 and recruit Rps31.

The authors have done a lot of work and the data presented is scientifically sound. Overall, I think that the data supports their growing model for the master regulator role of Ltv1 in the late stages of 40S head assembly and will be of significant interest to the field. I have a few comments for the authors to address for clarity:

Major Comments:

1. For the impact of Ltv1_L216S on 80S-like ribosome formation, would the authors explain why, in Figure 2B, the amount of 18S rRNA in the 80S fractions increases relative to wt Ltv1? Is that expected?

2. The authors compare doubling times for the mutant Ltv_S/A that abolishes phosphorylation to the Ltv1_L216S mutant, which also is not phosphorylated (Figure 2C). The difference in doubling time for the Ltv1_L216S mutant is very small compared to the difference for the Ltv_S/A mutant. Would the authors explain why the L216S mutation has only a small effect on growth when it also cannot be phosphorylated? Is there some other effect of the Ltv_S/A mutant or is there some level of phosphorylation of Ltv_L216S?

3. I think it is very interesting that the modeled Ltv1 structure is quite extended, spanning almost the entire head domain. I am wondering how to think about a single residue altering the network of interactions to globally effect the position of Ltv1, particularly since mutation of the neighboring residue has no growth defect (L217S). Can the authors comment on this? Have they modeled the L216S mutant protein to see if there are any predicted changes relative to wt? Also, based on their structure, would the authors comment on why L216S has an effect but L217S does not?

4. Interestingly, Rps9 is the only depleted body domain RP in the mass spec analysis of Ltv1_L216S ribosomes. The DMS data also show a cluster of altered reactivity near the Rps9 binding site. Do the authors think that this is real and significant? If so, would they comment on how Ltv1 influences Rps9 recruitment?

Minor comments:

1. In the introduction – Rps5 is referred to as uS2, but I think it is uS7.

2. Figure 2C – I think the Ltv1-L216C label for the right panel is missing.

3. In Figure 4F, should there be a significance comparison as noted in the legend? The authors state in the text that when the rRNA mutations are combined with Ltv1_L216S, there is a strong synthetic sick effect, but I am not sure which ones they are referring to. For example, the mutant 18S_C1327A appears to get better when combined with the Ltv1_L216S mutation.

4. Related to question 3, I am confused by the statement: “the rRNA mutations have little to no growth defect on their own (data not shown)”. Is this the data in Figure 4F as the blue bars with wt Ltv1? Can the authors also comment on the large distribution of doubling times for these mutants?

5. I think Figure S5 and 7 have some inconsistent labels and references in the text. In Figure 7, Ltv1_1-394 is referred to as Ltv1-C394 in the text. Figure S5B is labeled as Rps15 /Rps15_1-394, should that also be Ltv1_C394? Also, in the legend for S5B, it says Ltv1_3-394 but should be Ltv1_C394. S5E also has Ltv1_1-394 in the figure, should this be Ltv1_C394?

6. Figure 9A: what do the red and green lines mean? If it is the same as in B, I would suggest getting rid of the colors on part A because it is a little confusing when comparing direct interactions versus effects of Ltv1 on assembly of these proteins.

Reviewer #3: Previous work in the Karbstein lab showed that the ribosome assembly factor Ltv1 facilitates correct folding of critical helix junctions in the 18S rRNA. This report takes the story one step further by investigating the effect of a mutation in Ltv1 that maps to a region of human Ltv1 associated with a rare disease. An extensive set of mutations plus DMS structure probing convincingly show that a mutation in this region of yeast Ltv1 causes Ltv1 to bind pre-40S complexes improperly, preventing Ltv1 from facilitating assembly as it normally would. The structure probing results also led to a nice connection with Tsr1 movement.

I am not wholly convinced that the genetics experiments define the order of events as unambiguously as the authors propose (see comments below). However, the abundant data presented here clearly show that the Ltv1 mutation disrupts a very delicate set of RNA and protein interactions that are essential for normal 40S assembly. Overall, this study advances the current understanding of 40S ribosome assembly in yeast.

The experiments are thorough and well done. I have some comments related to the interpretation and presentation:

1. On pg. 4, human Ltv1 N168S is said to correspond directly to yeast Ltv1 L216S. However, this segment of Ltv1 is polar in mammals but hydrophobic in yeasts (Fig S1), and the sequence alignment seems tricky. Did the authors test other mutations in this region of Ltv1? It is clear from the results that the chosen mutation is consequential in yeast, but should one call it a “disease-associated mutation”?

2. In the introduction (pg. 3) the authors state that the hierarchy of ribosomal protein binding is difficult to explain because it cannot be accounted for by direct protein-protein interactions; it must depend on conformational changes that are induced by protein binding. The concept of induced fit is well established for bacterial ribosome assembly!! The authors may want to cite papers by Noller & coworkers PMID: 2459390, 3373531 or 2658053. The order of protein addition in yeast follows the bacterial 30S assembly map, only with the insertion of eukaryotic domain RPs. Perhaps this section of the introduction could be rephrased to better convey this point.

3. When interpreting the genetics results, the authors envision a defined sequence of molecular interactions that can be over-stabilized or blocked by mutations; as I understand it, these phenotypes are used to sequence the action of Ltv1. If any mutation were to totally block a particular step, however, pre-40S complexes would pile up and the strain would be dead. In reality, many of the mutations described here have mild effects on cell growth, suggesting there is some work-around. Can the authors comment on this?

4. The alphafold model of Ltv1 was impressively useful for interpreting and designing the experiments. I appreciate the effort to validate the predicted interactions. Nevertheless, “Inspection of the structure for Ltv1 on the nascent 40S revealed…” and like phrases imply an experimental structure model, not a computational one. The text and figure legends often state that the modeled Ltv1 complex “demonstrates” or “reveals” an interaction, which of course, it cannot. It can only predict. Please rephrase such claims wherever they occur.

5. On pg. 7, the rRNA mutations are said to have no strong effect on growth but a strong synthetic effect with Ltv1_L216S. In Fig. 4, some 18S mutations in fact cause a large spread in growth rates; can the authors comment? Is it possible that the strains are picking up some second site mutations? By contrast, the synthetic effect with L216S seems modest, and generally smaller for the 18S mutants than wt 18S, which I found surprising. Have I missed something here?

Sometimes it is easier to evaluate rRNA mutations by expressing a marked copy of the 18S from the plasmid so one can avoid shutting down the chromosomal rDNA. Did the authors consider this approach?

6. The authors conclude from the mass spec analysis that assembly of the head is generally perturbed, and I agree. In the next section, however, they interpret the relative abundance of Rps3, Asc1 and Rps20 as supporting the idea that Ltv1_L216S aberrantly stabilizes interactions between Rps3 and Rps20. I think the MS data are not precise enough to be interpreted with this level of detail because early assembly proteins (Rps9, Rps22, Rps5, Rps28, Rps16, Rps18) are depleted whereas other late assembly proteins (rpoS0) have normal stoichiometry. This goes against the idea of a specific block. This conclusion should be toned down a lot.

7. The data in Fig. 6 showing that addition of Rps29 (uS14) is impaired by Ltv1_L216S is very convincing. However, I had trouble following the logic of why Rps29 depletion should have a smaller effect in the Ltv1_L216S background than the wt background. If Ltv1_L216S tends to mask the binding site for Rps29, then low levels of Rps29 should make this even worse (synthetic), because the probability of Rps29 recruitment is reduced even further. Conversely, by this logic, Rps29 over-expression should rescue Ltv1_L216S. Since they observe the opposite, I wonder if the model is right. Perhaps I missed something?

Similarly, if Ltv1_L216S masks the Rps29 binding site by over-stabilizing Rps20-Rps3 interactions, then mutations that weaken the Rps20-Rps3 interface (Fig. 6D,E), or low RP levels (Fig. 6F,G), should rescue this effect rather than exacerbate it, which is not what the authors observed.

I agree that the results show that mispositioning of mutant Ltv1 results in assembly defects in the head, including impaired recruitment of Rps29, but the evidence for the sequence of events cartooned in Fig. 8 is weak, in my view. It would be helpful if the authors can better explain the logic (report actual doubling times?) or reconsider their model.

One alternative is that Ltv1 increases the rate of productive Rps29 recruitment and thus the likelihood that it will happen before Rps20 binds. This would not require Ltv1 to act directly on Rps20 and Rps3. Another model is that Ltv1 embraces Rps29, preventing its dissociation but also requiring Ltv1 to move to permit Rps20 and Rps3 binding. If the mutant is less able to respond to Rps20 or Rps3, it would explain why the Ltv1 mutant is less sensitive to low levels of Rps29 (less dissociation of this protein) but more sensitive to Rps20 and Rp3 mutations.

8. Please briefly indicate the methods used in the abstract (what type of data?).

9. I appreciate that the authors mention the systematic ribosomal protein names in the text. It would be even better if they would use these names throughout. The reason is that this will make their work more accessible to readers working on human or bacterial ribosome biogenesis. The other reason is that prefixes “u” or “e” in Figure 1 or Figure 8 will instantly convey that the key ribosomal proteins in this study are universally conserved and define the (ancient) architecture of the 40S head.

10. Pg 3, Rps5 is said to correspond to uS2; it should be uS7.

11. Fig. 3 legend, “Ltv1 separates Rps20_DE from Rps3_KK” – do the authors mean that Ltv1 lies between these two proteins?

12. Fig. 3 legend, “Ltv1 binds Rps3K75R76” – does this refer to association of Ltv1 and Rps3 proteins, or does it refer to a predicted molecular interaction between Ltv1 and these particular residues of Rps3? The latter is not usually referred to as binding, which typically implies a bimolecular association.

13. Fig. 7 & 8 – consider colors for Tsr1 early and late with higher contrast.

**Have all data underlying the figures and results presented in the manuscript been provided?**

Reviewer #1: Yes

Reviewer #2: Yes

Reviewer #3: Yes

PLOS authors have the option to publish the peer review history of their article (what does this mean?). If published, this will include your full peer review and any attached files.

Reviewer #1: No

Reviewer #2: **Yes: **Margaret Rodgers

Reviewer #3: No

---

## [Decision Letter · Decision Letter 1]

25 Sep 2023

Dear Dr. Karbstein,

Your revised version of "A disease associated mutant reveals how Ltv1 orchestrates RP assembly and rRNA folding of the small ribosomal subunit head" was evaluated be previous Reviewer 3. I am delighted to inform you that the reviewer concludes that you have addressed the concerns previously raised. There is now agreement from all of the reviewers and editors that your data and manuscript will make a fine contribution to PLOS Genetics. However, Reviewer 3 does raise one remaining issue concerning Fig. 9B (see below) which we believe you should be able to address now or in the proof. Therefore, your manuscript is editorially accepted for publication in PLOS Genetics. Congratulations!

Yours sincerely,

Anita K. Hopper

Academic Editor

PLOS Genetics

Gregory P. Copenhaver

Editor-in-Chief

PLOS Genetics

Comments from the reviewers (if applicable):

Reviewer's Responses to Questions

**Comments to the Authors:**

Reviewer #3: The authors have responded well to the critiques of their manuscript, and the revised manuscript is much improved. In particular, they have addressed concerns about the phenotypes of the 18S rRNA mutations (raised by both reviewers). The structural context for the genetics experiments is also presented more carefully in the revised manuscript than in the original. Overall this is an interesting and extensive study that advances the current understanding of the role of Ltv1 in yeast 40S biogenesis.

Although I am generally satisfied with the revisions, I gently point out to the authors that their model of the assembly pathway in Fig. 9B is structurally implausible. First, the topology of the contacts is cartooned incorrectly in Fig. 9B. 18S h39 does not lie between h41 and uS10 (Rps20) as diagrammed; uS10 binds at the interface of h39 and h41that are almost parallel with each other. uS14 (Rps29) has almost no contact with h39 or h41 – it binds h31 and the tip of h43. Second, uS10 (Rps20) truly does lie under the C-terminus of uS14 (Rps29), in the region of the Rps20_EYER mutations that define the interface between these two proteins. (The point of view is not relevant here.) Therefore, steric occlusion cannot be the reason why uS14 should bind first. However, because each protein mainly interacts with a separate domain of the 18S rRNA, not with each other, I agree that they could initially join the complex in either order. If uS14 binds first, its C-terminus presumably folds over the uS10 EYER residues after uS10 joins the complex.

I am convinced that the results here identify 40S interactions that depend on the Ltv1 chaperone. If the authors were to modify their cartoon in Fig. 9B to match the actual structure of the 40S ribosome, and perhaps acknowledge some ambiguity in the temporal sequence of events, that would be wonderful.

**Have all data underlying the figures and results presented in the manuscript been provided?**

Reviewer #3: Yes

PLOS authors have the option to publish the peer review history of their article (what does this mean?). If published, this will include your full peer review and any attached files.

Reviewer #3: No

**Data Deposition**

http://datadryad.org/submit?journalID=pgenetics&manu=PGENETICS-D-23-00751R1

**Press Queries**

---

## [Editor Report · Acceptance letter]

24 Oct 2023

PGENETICS-D-23-00751R1 

A disease associated mutant reveals how Ltv1 orchestrates RP assembly and rRNA folding of the small ribosomal subunit head 

Dear Dr Karbstein, 

We are pleased to inform you that your manuscript entitled "A disease associated mutant reveals how Ltv1 orchestrates RP assembly and rRNA folding of the small ribosomal subunit head" has been formally accepted for publication in PLOS Genetics! Your manuscript is now with our production department and you will be notified of the publication date in due course.

With kind regards,

Zsofi Zombor

PLOS Genetics

On behalf of:
